

# Copper permalloys for fluxgate magnetometer sensors

B. Barry Narod[1], David M. Miles[2]

[1]Department of Earth, Ocean and Atmospheric Sciences, University of British Columbia, Vancouver, Canada
[2]Department of Physics and Astronomy, University of Iowa, Iowa City, IA, USA

*Correspondence to*: B. Barry Narod (bbnarod@shaw.ca)

**Abstract.** Fluxgate magnetometers are commonly used to provide high-fidelity vector magnetic field measurements. The magnetic noise of the measurement is typically dominated by that intrinsic to a ferromagnetic core used to modulate (gate) the local field as part of the fluxgate sensing mechanism. A polycrystalline molybdenum-nickel-iron alloy (6.0-81.3 Mo Permalloy) has been used in fluxgates since the 1970s for its low magnetic noise. Guided by previous investigations of high

permeability copper-nickel-iron alloys, we investigate alternative materials for fluxgate sensing by examining the magnetic properties and fluxgate performance of that permalloy regime in the range 28-45%Cu by weight. Optimizing the alloy constituents within this regime enables us to create fluxgate cores with both lower noise and lower power consumption than equivalent cores based on the traditional molybdenum alloy. Racetrack geometry cores using six layers of ~30 mm long foil washers consistently yield magnetic noise around 4-5 pT/√Hz at 1 Hz and 6-7 pT/√Hz at 0.1 Hz meeting the 2012 1-second

INTERMAGNET standard of less than 10 pT/√Hz noise at 0.1 Hz.

## 1    Introduction

In December 2007 one of us (Narod) rediscovered a paper by von Auwers and Neumann (1935), titled in English "On Iron-Nickel-Copper Alloys of High Initial Permeability," and this eventually set into motion our examination of copper permalloys as potentially useful materials for fluxgate magnetometer sensors. Specifically, we are interested in copper permalloy's

potential to simultaneously provide low magnetic noise and low power consumption in a fluxgate sensing application. With the assistance of colleagues at Zentralanstalt für Meteorologie und Geodynamik [ZAMG, now Geosphere Austria] we had located a loose paper copy in a box of collected papers, situated in the library of the Austrian Academy of Sciences in Vienna, a collection which conveniently for us had been catalogued by their librarians. This paper was last cited in 1961 (Puzei, 1961), and had disappeared from living memory. A single citation of it in Bozorth (1951) had caused us to spend several years

searching for it. The collection of copper permalloy data included in von Auwers and Neumann (1935), extraordinary in both quantity and quality, are reproduced in translation in Appendix A.



Prior to our present efforts 6.0% Molybdenum (6%Mo, all %-compositions are given in weight-percent compositions.) permalloy was the state-of-the art sensor material for the best performing fluxgate sensors for geophysics and space physics (Gordon et al., 1968). However, the markets for such materials are small and in 1996 our community's ready access to such material ceased. Since that moment we and others have undertaken investigations of 6%Mo permalloy procurement and processing, to create new supplies (Müller et al., 1998; Narod, 2014; Miles et al., 2022).

We have now arrived at a place where we are able to melt in small quantities any permalloy, roll it, draw it, heat treat it and machine it to any desired shape, in any sequence of process steps. We began by fabricating 6%Mo permalloy (Narod 2014; Miles et al., 2016; Miles et al., 2022; Greene et al., 2022), and from it a variety of novel fluxgate sensors.

## 1.1    The case for copper permalloys

Miles et al., (2022) presented along with 6%Mo permalloy our first trial of a copper permalloy. This alloy consisted of 28% copper, 62% nickel and balance iron, which we designate 28Cu62Ni. The case for 28Cu62Ni went as follows:

We knew that 6%Mo permalloy has several properties that are thought advantageous for fluxgate sensor materials. These are 1) minimum magnetocrystalline anisotropy, $K_1$, 2) minimum bulk magnetoelastic anisotropy (magnetostriction) $\lambda_s$, 3) minimum saturation magnetization $\boldsymbol{B}_s$ of all such materials satisfying 1) and 2), and 4) a requirement for slow cooling during heat treatment, to minimize residual stress (Pfeifer, 1966; English and Chin, 1967; Pfeifer and Boll, 1969; Mussmann, 2010). Pfeifer (1966) and the subsequent papers all placed the zero-crossings satisfying 1) and 2) over a range of compositions including 4-6% molybdenum. But from their Figures 6%Mo uniquely also satisfied both 3) and 4), and these papers' Figures ultimately drove the choice to use 6%Mo in a new generation of fluxgate magnetometers (M. Acuña, 1981, personal communication).

We then sought a copper alloy composition that satisfied four equivalent conditions, a choice which was enabled by data presented in the von Auwers and Neumann (1935) paper. Their Figures 8, 11, 13, included here in Appendix A, respectively plot contours of initial permeability, magnetostriction and saturation magnetization for a collection of measurements from 130 copper permalloys, all specimens having undergone the authors' "1100-treatment", a slow-cooled heat treatment.

We consider the 1100-treatment to be functionally equivalent to that of the slow-cooling of Pfeifer (1966). We have also taken the highest initial permeability to be a proxy for minimum total anisotropy, both magnetocrystalline and magnetoelastic.

For their "permalloy-treatment", essentially quenching, we noted in their Figure 6 that maximum initial permeability locates at a composition including 14%Cu, while for 1100-treatment the maximum locates at a composition including 28%Cu. The



saturation magnetizations, $B_s$, for the two cases were respectively 750 mT and 550 mT. Our condition 3) then led us to
28Cu62Ni as our first copper test alloy which conveniently would share the same heat treatment as 6%Mo permalloy.

Our first trial with a fluxgate sensor made from 28Cu62Ni (Miles et al., 2022) produced performance results comparable to a nominally identical sensor made from 6%Mo permalloy, and for some parameters such as power consumption the 28Cu62Ni alloy sensor outperformed. The success of our initial copper alloy trial encouraged us to expand our investigations of copper permalloys, seeking more resolution in composition and extending our fluxgate sensor builds to alloys with much higher copper contents. It is well known that lower $B_s$ leads to improved noise performance (Mussmann, 2010), and Appendix A, Fig. 13 shows clearly that higher copper content reduces $B_s$, but higher copper content also worsens initial permeability, our proxy for anisotropy. With respect to noise the theory of Narod (2014, Eq. 8) suggests that the effect of lower $B_s$ could dominate over lower permeability.

The choices for our alloy compositions were again guided by Figures 8, 11, 13, of von Auwers and Neumann (1935). For the copper contents we selected the range 28-45%Cu. For our minimum copper-content alloys we chose 28%Cu with about 60%Ni. This coincided with the maximum initial permeability measured by von Auwers and Neumann (1935). For our maximum copper-content alloy we chose 45%Cu, 50%Ni, 5%Fe which has a saturation induction of about 200 mT. We were concerned we might be running into a Curie temperature limitation; however, this has now been determined not to be the case. For the nickel contents we selected a range of 5%Ni content centred on the zero magnetostriction contour. On the copper and nickel axes our specimen spacings were typically 2% and 1% respectively. This collection included 52 specimens, with the composition range plotted in Fig. 1.



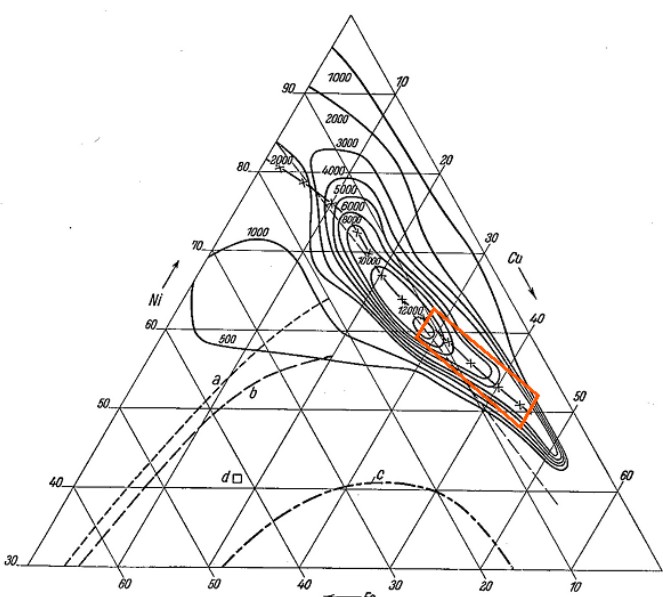

**Fig. 1. The range of copper-permalloy compositions in our study, outlined in red and superimposed on von Auwers and Neumann (1935) Fig. 8, their contour plot of initial permeabilities after their 1100 treatment.**

The organization of the remainder of this paper is as follows: We first offer a look at the history of permalloy development, beginning in the 1920's, with our attempt to answer a question: How is it possible that copper permalloys for use in fluxgate magnetometers was missed for most of a century? We then continue with descriptions of our experimental work including sample preparation and data acquisition, and the presentation of our data. We conclude with our interpretation of our data.

## 1.2     History of Permalloys as soft magnetic materials

We have asked ourselves how was it possible for high copper-content permalloys to have remained undiscovered for nearly a century, for their use as suitable sensor materials for low noise, low power fluxgate magnetometer sensors? Here we examine the history of permalloy development and attempt to consider this question. We loosely use the word permalloy to mean a low loss, low coercivity alloy of more than 35% nickel, with some iron content (Nature, 1948). Other alloying elements may include chromium, molybdenum, or copper.

Our earliest period of interest began in 1921 with Gustav W. Elmen's filing for US Patent 1,586,884 while employed by the Western Electric Co. of New York (Elmen, 1926), later, Bell Telephone Laboratories. This period concluded in 1947 with the development of 'Supermalloy' (Boothby and Bozorth 1947), also at Bell Telephone Laboratories. This interval features the invention of the fluxgate magnetometer by Hans Aschenbrenner and Georg Goubau (1936), and the development of the Gulf magnetometer in 1941 (Vacquier, 1946a,b; Wyckoff, 1948). The interval also included the development for The Telegraph

Construction and Maintenance Co. Ltd. in the United Kingdom by Willoughby Statham Smith and Henry Joseph Garnett of





early copper permalloys (Smith and Garnett, 1925,1926,1927), both investigating developments for magnetic loading of communications cables. In the 1930's Otto von Auwers and Hans Neumann, both at Siemens & Halske in Berlin, investigated a range of ternary iron-nickel-copper alloys, for both inductive and permanent magnet properties (von Auwers and Neumann, 1935; Neumann, 1934; Neumann et al., 1937). We shall now look more closely at the US, UK, and Germany efforts.

### 1.2.1 USA

Elmen's 1921 patent filing claimed an alloy comprising 78.5% nickel / 21.5% iron, the first higher-permeability nickel-iron alloy. Elmen noted within the patent text the inverse relation between minimum hysteresis losses and maximum permeability in the vicinity of the target composition, the presence of a resistivity maximum at mid-range compositions above the Invar transition, and the presence of the "well known maximum ... that precedes the attainment of the critical temperature" attributable to Hopkinson (1889). In 1926 Elmen filed his second permalloy patent application (US patent 1,768,443, 1930), claiming a lower loss, lower coercivity alloy of about 78.5% nickel, 16%-18.5% iron and 3%-4% molybdenum. The motivation for Elmen's efforts was the creation of low loss magnetic materials for the magnetic loading of central-conductor trans-ocean communications cables, with the intent of increasing their signalling speed (Foster, 1928). Elmen noted in the 1930 patent that 3.7% molybdenum content developed for that time the highest initial permeability of any known magnetic material. The importance of the alloy's heat treatment was to some extent understood by Elmen. A typical heat treatment features a reducing atmosphere, a period held at a higher temperature, e.g., 1100C (the soak), and a controlled cooling from 600C to room temperature. Extrema to the cooling could be a quench where a 600C work piece was removed from heat and placed on a copper slab, or a furnace cool when a furnace was allowed to slow cool to room temperature, taking perhaps 15 hours to complete. Elmen tested both types referring to them as "A - slow cooling", and "B - rapid cooling".

Twenty years later Boothby and Bozorth (1947) reported on their lab's highest yet initial permeability in a 5% molybdenum permalloy, which they named "Supermalloy". Unlike prior materials their new alloy benefited from "a definite cooling rate to be used below the temperature at which atomic ordering begins", without disclosing the rate, which likely was in the vicinity of 200C/hour (Carpenter Technology, 1991). Their hypothesis was that the addition of only molybdenum to the binary nickel-iron alloy together with a particular cooling rate caused magnetoelastic anisotropy (magnetostriction) and magnetocrystalline anisotropy to "disappear at the same time".

In subsequent years both Elmen and Bozorth participated in investigations at the magnetics laboratory of the US Naval Ordnance Laboratory. It appears at all times these US efforts to develop permalloy materials concentrated on only ternary molybdenum permalloys.



### 1.2.2    United Kingdom

Smith and Garnett obtained three patents with respect to copper permalloys for use in communications cables (Smith and Garnett,1925,1926,1927). Their development of 5% copper permalloy enabled them to circumvent the molybdenum patents of Elmen. The first two patents could be considered together as a single publication as they have identical filing dates and almost identical content. There are differences between their claims. The 1925 patent claims compositions including 15%-25% copper. The 1926 patent claims compositions including 5%-6% copper. In both patents' claims the iron content is never

less than 17%. The 1927 patent refers primarily to the heat treatment of magnetically loaded wires and makes no claims with respect to compositions. That said this patent does refer to one composition: 76% Nickel, 16.5% Iron, 5% Copper, 2.0% Chromium, 0.5% Manganese which became the historical composition of "Mumetal". Smith and Garnett in these patents do not describe a role for the copper content, but they very clearly ascribe a role for a fourth element, typically a transition metal, vanadium through to chromium, or molybdenum, that role being to increase the resistivity of the alloy.

The US investigators and the United Kingdom investigators thus had very different understandings regarding the role of molybdenum in ternary or quaternary permalloys. What follows now should demonstrate that in Germany the Siemens & Halske investigators were thinking in line with the United Kingdom investigators.

### 1.2.3    Germany

Similar investigations were undertaken in the 1930's in Berlin at Siemens & Halske Wernerwerkes. These efforts led to the

development of their commercial alloy "1040", also known as "M-1040" (Pfeifer and Boll, 1969). "1040"s composition is 72% nickel, 11% iron, 14% copper, and 3% molybdenum (Neumann 1934). Neumann specifically states its resistivity as "0.56" (presumably 56 microohm-cm), which is competitive with modern molybdenum-permalloy resistivities, and significantly higher than our own measurements of 28% copper-permalloy resistivity near 31 microohm-cm.

Comparing magnetic values for "1040" with those for ternary 14% copper-permalloy (von Auwers and Neumann, 1935) the

addition of 3% molybdenum reduced saturation magnetization from about 800 mT to 600 mT, a significant reduction, and similar to that for a modern molybdenum permalloy with saturation magnetization typically about 0.75T. Whether the Siemens & Halske investigators understood the full role of molybdenum content in permalloy is unclear. They certainly appreciated its role in reducing conductance losses and would have noted its effect on saturation magnetization. Our own investigations replicating "1040" found one more issue. Our "1040" specimen developed significant chemical segregation when cooled

slowly, presumably due to its quaternary composition. Enoch and Murrell (1969) also conjectured such segregation in quaternary permalloys. Thus like 14% ternary copper-permalloy "1040" likely developed its best magnetic properties when it underwent "permalloy-handling", that is, air-quenching from 625C (von Auwers and Neumann, 1935).



Like Elmen, von Auwers' and Neumann's 1935 broad-based study of 130 ternary iron-nickel-copper alloys also considered two distinct heat treatments, one rapid and one slow. As noted above, a 14% copper-permalloy achieved its best performance

for the rapid cool, with respect to initial permeability. For their furnace-cooled specimens their highest initial permeability was achieved at 28% copper, the composition also used for our first copper-permalloy test (Miles, 2022).

Georg Keinath (Melz, 1960) was in this period the head of Siemens & Halske's entire development department of the Wernerwerkes for measuring technology. In 1937 he and his Jewish wife emigrated to the USA. Keinath's departure from Berlin coincided with Siemens & Halske's diminution or cessation of permalloy research in general and of copper-permalloy

research in particular.

### 1.2.4    Chromium and molybdenum

Keinath (1932) presented a study of the effects of chromium and molybdenum in ternary permalloys, He looked particularly at their effects on resistivity, Curie temperature, and saturation moment. Measured by weight percentage, chromium produced the larger effect, affecting resistivity by about 10% in excess of molybdenum's effect, and for saturation moment almost

doubling that of molybdenum. For 3.7% chromium content the saturation moment compared to binary permalloy, was reduced from 1.0T to below 800 mT. Curie temperature went from 580C for binary permalloy, to 450C for 3.7%Mo-permalloy, to 230C for 10%Mo-permalloy to -20C for 15%Mo-permalloy.

Farcas' (1937) data plotted by Bozorth (1951), Fig. 10-11, shows a strong dilution effect of chromium upon cobalt. Narod (2014) from a variety of data sources found that one molybdenum atom was able to cancel those effects from almost eight

nickel atoms, or two iron atoms, a result broadly in agreement with the approaches of Enoch and Fudge (1966), and Enoch and Murrell (1969).

Considering atomic percentages as contrasted with weight percentages the effects of chromium and molybdenum on saturation moment on a per atom basis are nearly identical. With respect to resistivity on a per atom basis the effect of molybdenum is noticeably larger than that of chromium. Today molybdenum dominates within the ternary permalloys. Chromium may also

have fallen out of favour in part due to its volatility at high temperatures in the presence of hydrogen, a trait we have observed directly.

### 1.2.5    The second Interval (1953-1969*)*

In the 1960's another look was taken at these alloys, with regard to better understanding magnetic anisotropies, and also specifically looking for improved magnetic materials for fluxgate magnetometer sensors (Bozorth, 1953; Puzei and Molotitov,

1958; Puzei, 1961; Puzei, 1962; Odani, 1964; Lykens, 1966; Scanlon, 1966; Pfeifer, 1966; Odani and Sunazawa, 1967; Cohen, 1967; Scholefield et al., 1967; Snee, 1967; Gordon et al., 1968; English and Chin, 1969; Pfeifer and Boll, 1969). This activity



covered the planet, from Nippon Telegraph and Telephone in Japan, Telcon Metals Ltd. in the United Kingdom, Vacuumschmelze in Germany, the Central Research Institute of Ferrous Metallurgy in the former Soviet Union, and in the USA, Bell Telephone Laboratories, Carpenter Steel Co., Allegheny Ludlum Steel Corp., Magnetics Inc., and the U.S. Naval Ordnance Laboratory.

With respect to fluxgate sensors, this led to Gordon et al.'s (1968) well known 6.0% molybdenum permalloy, a slow-cooled material. 14% copper-permalloy was still in human memory at this time (Pfeifer and Boll, 1969) but it appears that the slow-cooled 28% copper-permalloy had been ignored or forgotten, including in the 1951 book "Ferromagnetism" (Bozorth, 1951).

The role of molybdenum was described in detail by Pfeifer (1966), after data from Puzie (1962) and Puzie and Molotitov (1958). His Fig. 5 presents contours of zero magnetostriction, ($\lambda_s = 0$, $\lambda_{111} = 0$, $\lambda_{100} = 0$), and zero magnetocrystalline anisotropy ($K_1 = 0$) for various cooling rates including oil quenching and very slow cooling, and notes the two coincided for rapid cooling at about 4%Mo, and for very slow cooling at a little over 6%Mo.

### 1.2.6    1969 – present

For the purpose of producing improved fluxgate magnetometers in quantity, in 1969 the US Naval Ordnance Laboratory commissioned an ingot of the 6.0%Mo, 81.3%Ni, Fe balance alloy (6Mo81Ni), and had it rolled to 12.5µm foil (M. Acuña, 1981, personal communication). We conjecture that the ingot and foil were produced by the Hamilton Watch Company and their then metallurgical division, later spun off as Hamilton Precision Metals. Eventually this material went to Infinetics Inc., then a producer of ringcores, where they created their S1000 6-81 ringcore, which from 1979 to 1996 were provided to fluxgate magnetometer developers, going end-of-line for lack of alloy foil. [In the 1990's Müller et al., (1998) in Germany also investigated 6%Mo permalloys but with significantly different results and lower yields (H. U. Auster, personal communication, June 12, 2008), possibly due to significant constraints on their available heat treatments (K. H. Fornacon, personal communication, April 19, 2017).]

The success of the Naval Ordnance Laboratory 6.0% molybdenum-permalloy as a fluxgate material (Gordon et al., 1968) effectively obviated the immediate need for seeking other novel materials. Then as now, 4-5% molybdenum permalloys are also used for fluxgate sensors. It was not until the introduction of amorphous alloys to fluxgates (Shirae, 1984, Narod et al., 1985) that any newer materials were added to the fluxgate sensor roster. These authors noted strong correlations between fluxgate noise with respectively, Curie temperature and saturation moment, $\boldsymbol{B}_s$. Musmann (2010) has more recently pointed out that this relationship has become common knowledge.

More recent permalloy studies have looked at crystallographic influences on performance rather than chemistries, for example: Major and Martin, (1970); Couderchon et al., (1989); Herzer (1990); Herzer, (1993); Müller et al., (1998). All have found that



grain alignment or size have significant effects on magnetic properties. Anticipating this direction, Odani (1964), (Odani and Sunazawa,1967) was an earlier investigator to consider recrystallization, both primary and secondary, to have a role in determining magnetic material performance, in parallel with the effect of alloy compositions.

None of these relatively recent studies considered copper as suitable content for permalloy. 14%Cu-permalloy was likely
abandoned due to its lower resistivity, which would lower its usable bandwidth (Chaston, 1936). 28%Cu-permalloy was never considered as being suitable for anything. 6%Mo-permalloy never entered the mainstream for magnetic materials due to its higher material cost, lower saturation moment and awkward optimal heat treatment (slow cooled).

Puzei (1961) was the last investigator to look at ternary copper-permalloys. A copy of the only publication with extensive copper-permalloy soft magnetic data (von Auwers and Neumann, 1935) proved very difficult to locate, its only citations being
in Chaston (1936), Bozorth (1951) and Puzei (1961). To address this shortfall, we offer an English language translation of it, in Appendix A.

## 2 Sample preparations

Part of our acquired data were intended to fill in measured parameters presented in von Auwers and Neumann (1935). These include room temperature saturation magnetization, initial permeability, coercivity and magnetostriction. Our additional
measures include DC resistivity, and for four selected compositions we acquired $B/H$ curves across a range of temperatures, up to their Curie temperatures. We also used these same four compositions in fluxgate sensor builds. Here we describe in detail our specimens' production.

### 2.1 Alloy manufacturing

For our investigations we have planned to produce up to seventy individual Cu-Ni-Fe alloy specimens, with copper weight
contents from 28% to 50%. Here we report our initial phase in which we have produced fifty-two and tested fifty alloys with copper contents from 28% to 45%. The alloys were produced as batches of fifteen 30g circular ingots, melted at 1550C from high purity powders, each in a covered ceramic crucible. Two ingots failed during cold rolling. The full procedure is described in Miles et al., (2022).

We then made three experimental specimens from each alloy, one each for magnetostriction, DC resistivity, and $B/H$
explorations. After metal fabrication, all specimens received a common heat treatment featuring a four-hour soak at 1125C, and a slow cool (35C/hour) from 600C to room temperature. Details regarding the heat treatment are also available in Miles et al., (2022). [For comparison, the 1935 Siemens study used 7kg melts, (von Auwers and Neumann, 1935).]



### 2.2 Magnetostriction specimens

Magnetostriction specimens were the first to be produced from the alloys. We cold rolled each ingot to a thickness of 1.00mm,
then cut a rectangular specimen 8.0mm x 50.0mm. Each specimen received a stamped four-digit numerical code to reflect its
identity, a marking that would survive the heat treatment. We polished one surface of each specimen to better reveal the grain
structure after heat treatment. Each specimen received a MgO dipped coating for isolation during the heat treatment. After heat
treatment each specimen received a 12mm x 3mm strain gauge, attached using cyanoacrylate adhesive. Magnetostriction
measurements have now been left for a future investigation. [For comparison, the 1935 Siemens study used strips 0.35mm
thickness, 10mm width, 100mm length, with less than half the cross-section of our present specimens]

### 2.3 Resistivity specimens

The resistivity and $B/H$ specimens both utilized permalloy strips of nominally 0.100mm x 3.20mm cross section. 2g remnants
from the 1mm sheets got further cold rolled to thickness 0.100mm, then slit to 3.2mm strips. We selected one such strip each
for the resistivity and $B/H$ specimens.

We again polished part of the first strip surface, stamped a two-digit identity mark into the strip, and coated it with a MgO dip.
80mm was the minimum length used for the resistivity measurement. We measured strip thicknesses, widths, lengths, and
masses prior to the heat treatments. Together these data provide some redundancy, which we used for estimating errors. We
estimate dimensional and mass rms error for resistivity at 0.2%. Curiously von Auwers and Neumann (1935) did not include
DC resistivity measurements although it was well known in their time that copper alloy resistivities could be undesirably low
(Smith and Garrett, 1925,1926; Neumann, 1934).

### 2.4 $B/H$ specimens

For each $B/H$ specimen we produced a bobbin machined from a Sch80 ¾" [20mm] pipe [nominal O.D. 26.67mm], grooved
to receive the permalloy strip, groove diameter about 23.5mm. The bobbin material chosen for good thermal match was 70%
cupronickel, UNS C71500. Bobbins were stamped with two-digit identity codes to match those for the resistivity specimens.
Again, dimensions and masses were measured for each bobbin assembly to reduce $B/H$ errors introduced by such
inaccuracies, and again we estimate such errors at 0.2% rms. In all cases mass measurements were rationalized with
dimensional measurements using alloy densities estimated by elemental mixing. We estimated magnetic path length from the
mass and cross-section area data.

After heat treatment each ring received twenty-turn, color-coded bifilar windings. Foil overlaps for spot welding likely
produced the largest uncertainties which we estimate created typical errors of about 1% in all magnetic measurements. [For



comparison, the 1935 Siemens study used punched circular rings in groups of ten, 0.35mm thickness, 45mm inside diameter, 60mm outside diameter.]

### 2.5    Curie temperature specimens

Of the fifty alloys we tested, fourteen were selected for possible magnetic properties studies at elevated temperatures, up to
their respective Curie temperatures. These rings received windings of AWG26 [400 micron] single-strand bare-copper, insulated with color-coded, FBGS series, braided-fiberglass sleeving (Omega Sensing Solutions, 2022). This insulation has a rating of 616C for short durations.

### 2.6    Vibrating sample magnetometer specimens [VSM]

A few alloys received further cold rolling to 0.050mm thickness. We have cut 3mm discs from these and flattened them prior
to heat treatment. Our plan is to subject these to a more careful $B/H$ examinations via VSM, and x-ray diffraction examinations seeking recrystallization fabrics such as those found by Major and Martin, (1970).

### 3    Data acquisition

We collected resistivity and $B/H$ data for all specimens. We summarize all our test data on composition-based grids laid out as shown in Fig. 2.

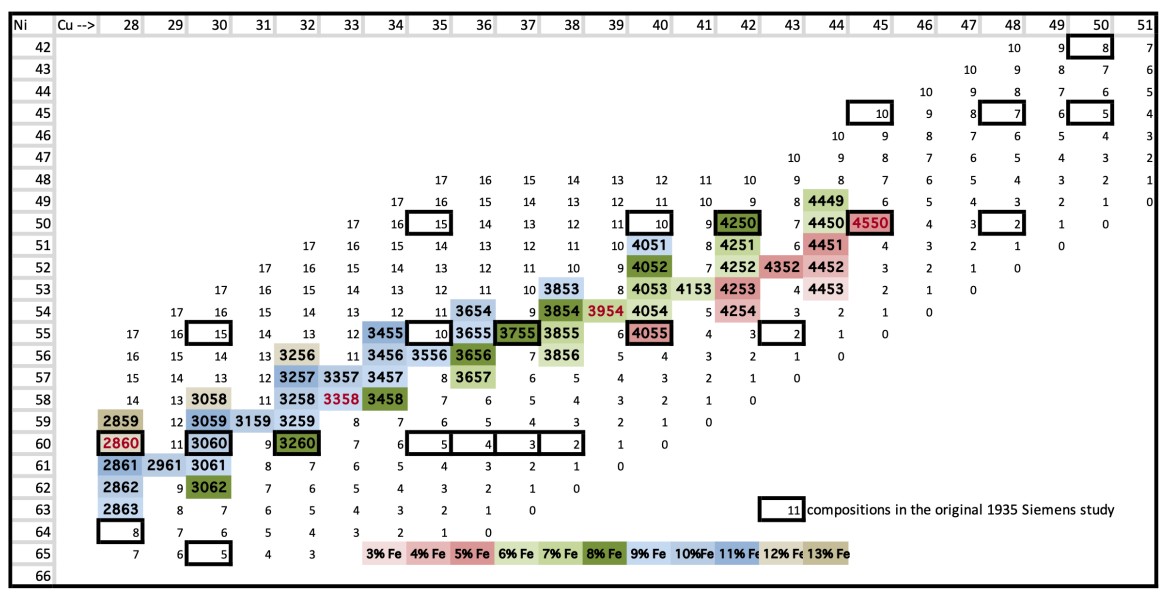


**Fig. 2. Compositions of the fifty-two copper-permalloy specimens.**



Here copper content is plotted in 1% increments, increasing to the right. Nickel content is also plotted in 1% increments, increasing down. Our physical specimens' cells are all marked in colour. Each cell composition then has a four-digit code indicating the alloy content with the first two digits stating copper content, and the last two digits indicating nickel content.

Iron content is the remainder, indicated in the un-used cells Fig. 2. Here the colour code indicates iron content with low iron contents in reds located to the right, and high iron contents in blues located to the left. Cells with black outlines are compositions included in the von Auwers and Neumann (1935) study.

Four cells presented with red lettering codes, namely 2860, 3558, 3954 and 4550 are our specimens also used in Curie temperature tests and in fluxgate sensor builds.

**3.1 Resistivity**

To measure resistivity each strip specimen was driven with a 2.00A DC current, and pin probes with either 80mm or 100mm separation completed a four-contact resistance measurement. Resistivity data are presented in Fig. 3. Highest values occur for highest copper contents, where nickel contents are similar. This is a well-known effect associated with such alloy mixing. Higher resistivities are also associated with higher nickel contents for similar copper contents. Both effects are to our advantage

and will be discussed further.

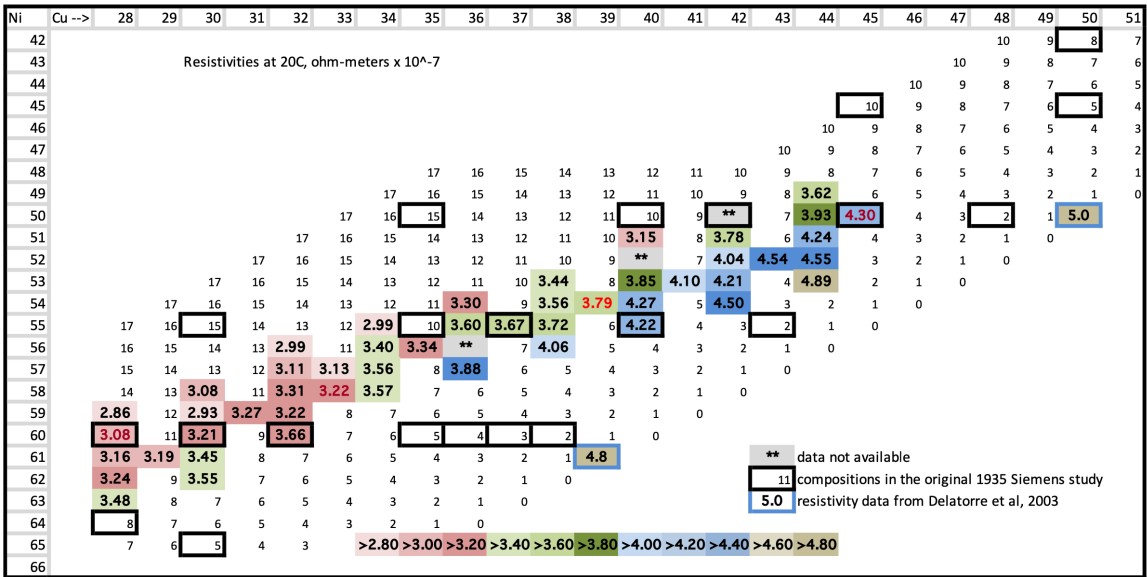

**Fig. 3. Resistivity data, in units of ohm-metres x10⁻⁷. Colour coding indicates lowest resistivities in reds, to the left, and highest resistivities in blues, to the right.**



### 3.2 Saturation induction

Saturation induction values range from 600 mT for a 28Cu60Ni specimen down to below 200 mT for the highest copper content specimens. Our values are consistent with those presented in von Auwers and Neumann, (1935). Subject to error limits, saturation induction consistently decreases with either increasing Ni or increasing Cu content. One high-copper-content, high-nickel-content specimen, namely 4453, is paramagnetic at room temperature. Data are presented in Fig. 4.

Test conditions for saturation inductions were magnetic field amplitude ±1000 A/m, triangle field waveform at 24Hz. These fields were generated using a generic signal generator, one of the 20-turn windings and an Accel Instruments TS250-0 waveform amplifier. Data were acquired from the second 20-turn winding using an operational amplifier integrator and a 12-bit PicoScope as our digitizer.

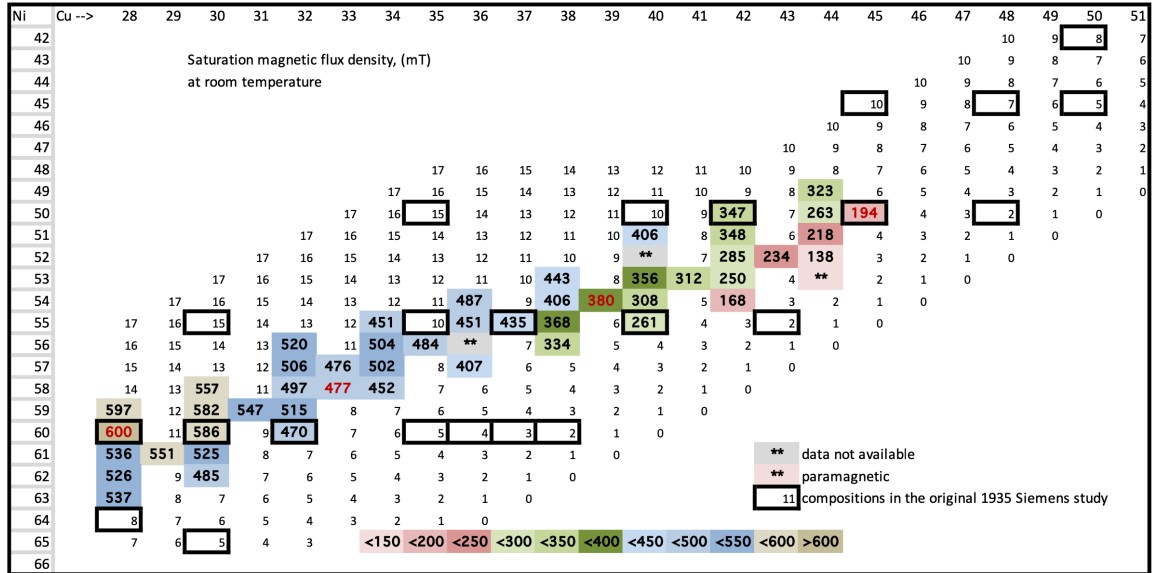

**Fig. 4. Saturation induction, $B_s$, in mT. Highest values are presented in blues, to the left, and lowest values are presented in reds, to the right.**





## 3.3    Curie temperatures

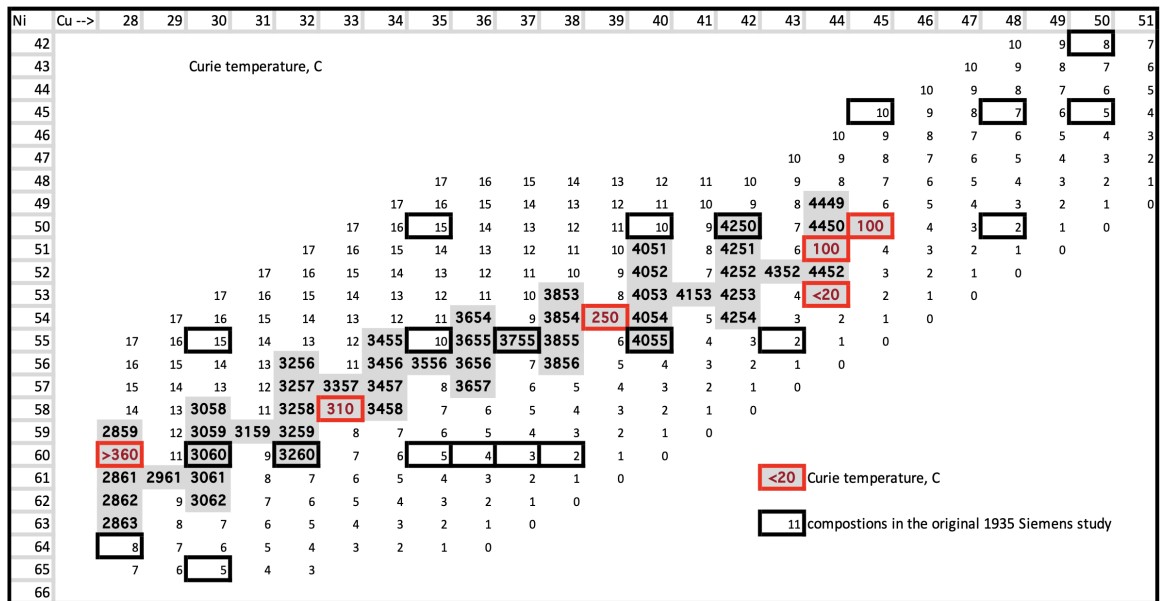

**Fig. 5. Curie temperatures. We have six specimens with Curie temperature estimates. These specimens are indicated as red-outlined cells, with Curie temperature Celsius estimates marked with red lettering therein.**

We estimated Curie temperatures ($T_c$) for our selected specimens by raising their temperature and estimating when saturation induction went to zero. All specimens were contained within an airtight quartz glass cylinder with suitable electrical feedthroughs. We heated the specimens under argon gas up to 360C, so as not to risk affecting their magnetic properties. Data are presented in Fig. 5.

As expected, alloy 28Cu60Ni had the highest $T_c$, over 360C, somewhat higher than that we measured for molybdenum

permalloy, 6-81 [360C]. Alloys 45Cu50Ni and 44Cu51Ni both measured at about 100C making them potentially useful in magnetometers. Alloy 44Cu53Ni being paramagnetic received a $T_c$ estimate, <20C.

## 3.4    Initial permeability (relative)

Test conditions for initial permeabilities were magnetic field amplitude ±0.4A/m, triangle waveform at 240Hz. Our results are presented in Fig. 6.






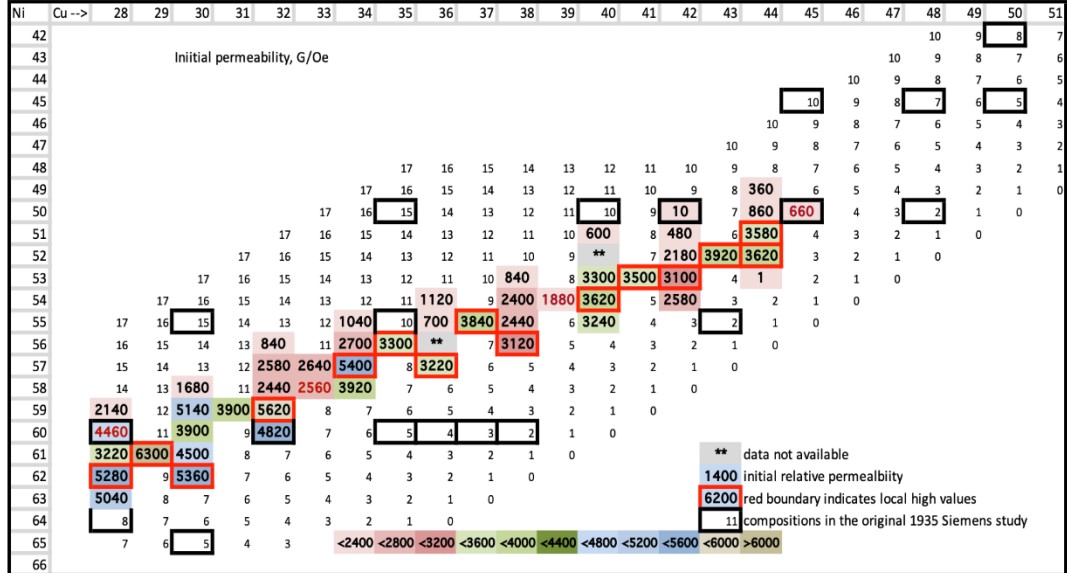

**Fig. 6. Initial permeabilities. Highest permeabilities are colour-coded as blues and browns. Fifteen cells have been outlined in red with which we are indicating local high values for initial permeabilities.**

In Fig. 6 we've selected fifteen cells to highlight with red outlines. These are local high values and decrease to the right. This overall decline in permeability values with increasing copper content is slight, not as high as for saturation induction. For

fluxgate magnetometers this effect should be beneficial. With increasing copper content, while saturation induction is declining (beneficial) the other predicted key magnetic property, anisotropy, as reflected in permeability is not degrading rapidly, if at all. In the next section on coercivity the same fifteen red outlines are repeated in Fig. 7.

### 3.5    Coercivity

The test conditions for coercivity were magnetic field amplitude ±50A/m, triangle waveform at 10Hz. We increased the sensing

windings to 84 turns. Our results are presented in Fig. 7. For the fifteen marked cells coercivity ranges from a high of 7.5 A/m for the lowest copper content specimens, down to 2.5 A/m for the highest copper contents. These values are comparable to those of amorphous alloys favoured by fluxgate developers, e.g., VITROVAC 6025, (Sekels, 2019).



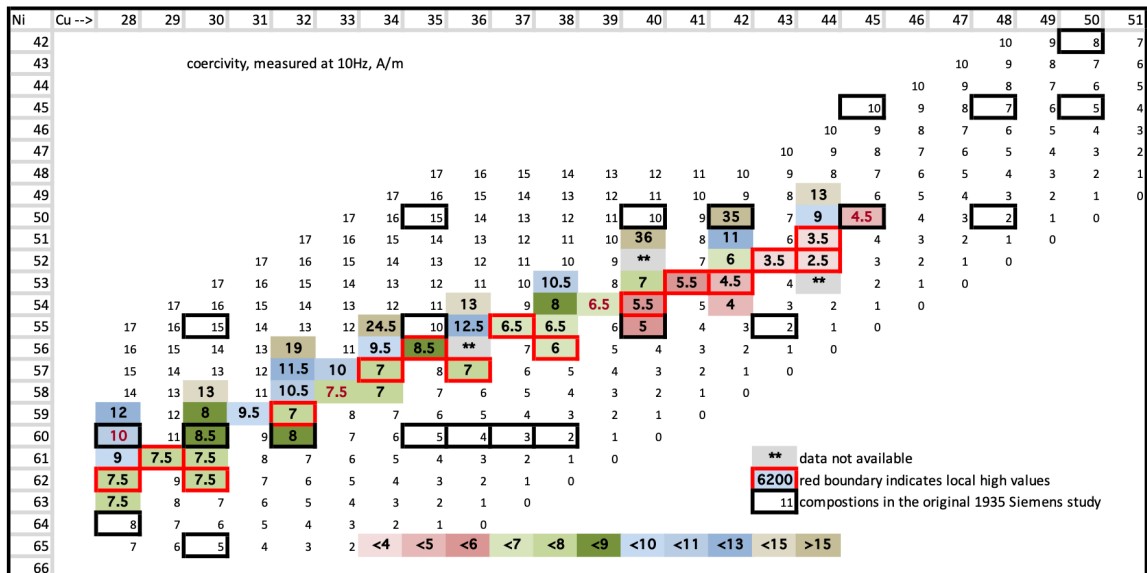

**Fig. 7. Low frequency coercivities. Highest coercivities are colour-coded as blues and browns. Fifteen cells have been outlined in red with which we are indicating local low values for coercivity. Values are in units of A/m.**

The fifteen locally low value coercivities correspond almost perfectly to the locally high permeability values in Fig. 6. We believe these two measurements are exploring the same phenomenon – magnetic anisotropy's influence on domain wall motion. For permeability, the measurements examine reversible domain wall motion. For coercivity, the measurements examine the endpoint of reversible domain wall motion and the entry into irreversible domain wall motion. In both cases, permeability and coercivity, higher domain wall mobility improves our measurements' results.

### 3.6    Fluxgate tests

Alloys 28Cu60Ni, 33Cu58Ni, 39Cu54Ni and 45Cu50Ni were fabricated into racetrack fluxgate sensors of the same design as in Miles et al., (2022), three sensor cores per alloy. We included sensors incorporating alloy 6.0%Mo,81.3%Ni (6Mo81Ni), for comparisons. Each sensor core included three 50-micron racetrack foils.

Our fluxgate testing results for five alloys are presented in Fig. 8. For each of the fifteen sensors we have noise power spectral densities [PSD] at both 1.0Hz and 0.10Hz, and core drive power consumption. In all cases power consumption for a copper alloy sensor is lower than power consumption for any 6-81 sensor. Generally copper alloy sensors have lower noise PSD at both frequencies, and significantly so for 45Cu50Ni sensors which have 1.0Hz noise PSD values within the range 6-8 pT/√Hz.



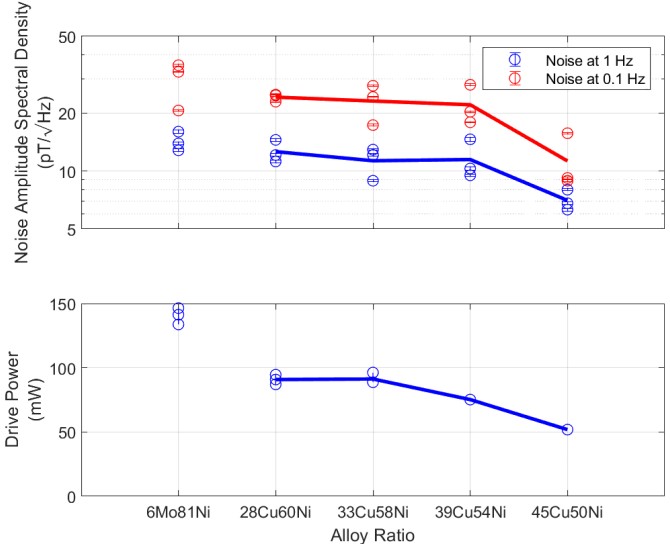

**Fig. 8. Fluxgate noise PSD and power consumption for five alloys.**

We further examined the performance of alloy 45Cu50Ni by producing twenty-five 6-layer racetrack core sensors. Noise PSD results for these sensors are presented in Fig. 9.

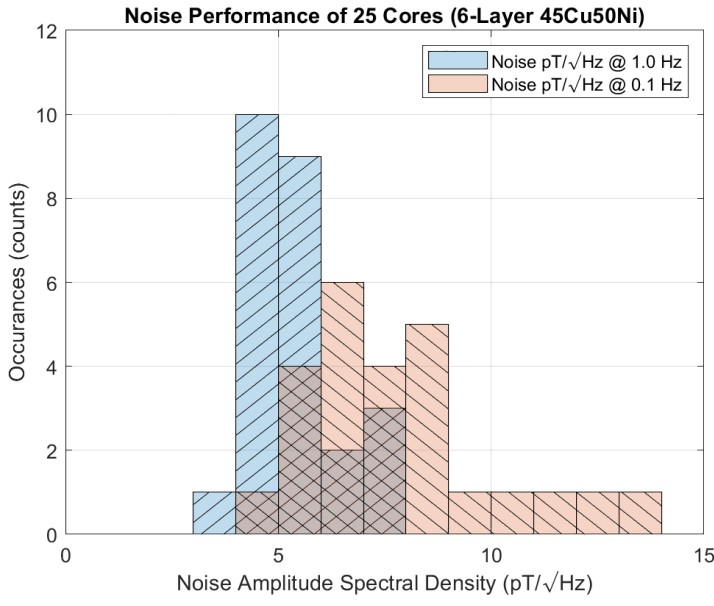

**Fig. 9. 1.0Hz and 0.10Hz noise PSDs probability density plot for twenty-five 6-layer 45Cu50Ni racetrack sensor cores. Cross-hatched columns indicate sensor cores for both test frequencies.**



Here we found 1.0Hz noise PSDs in the range 3-8 pT/√Hz and most likely in the range 4-5 pT/√Hz. 0.10Hz noise PSDs are in the range 4-14 pT/√Hz, peaking at 6-7 pT/√Hz. Of the twenty-five cores, twenty-one have 0.10Hz noise PSD below 10 pT/√Hz, that is the noise target for INTERMAGNET magnetic observatory variometers (Turbitt, 2014). These cores are intended for use in pairs in our TESSERACT sensors (Greene et al., 2022).

## 360  4     Discussion

The concept for our investigation has been to fill in magnetic properties data within the range of copper-permalloys presented in Fig. 1, and to test fluxgate magnetometer performance for a small selection of compositions within that range.

Regarding the magnetic properties data, we have replicated the results of von Auwers and Neumann (1935) and improved chemical content resolution. We discuss here our results in the order in which we have presented our data.

## 365  4.1     Resistivity

Resistivity has been investigated for binary copper-nickel alloys but has to our knowledge not been previously investigated for ternary iron-nickel-copper alloys. For copper-nickels, resistivity peaks at about 45%Cu (Copper Development Association Inc, 2023). Our data broadly replicate those early results. From 28%Cu to 45%Cu resistivity increases about 2% for every 1% increase in copper. In addition, we see a roughly 8% increase in resistivity for every 1% increase in nickel content. For our 370 lowest noise tested alloy, 45Cu50Ni, we measured resistivity 4.30x10$^{-7}$ohm-m, of similar magnitude to Mo-permalloys. From our measurements we have concluded that resistivities for our alloys of interest are not a limitation.

## 4.2     Saturation induction

Our data closely replicate results in von Auwers and Neumann (1935). Because saturation induction varies only slowly with alloy composition our results have only filled in predictable details.

## 375  4.3     Curie temperatures ($T_c$)

Prior to going into our 50-alloy study we knew that 28Cu60Ni would be a usable alloy for magnetic sensors. Von Auwers and Neumann (1935) had viable saturation data with as high as 50% copper, but in order to determine what could be a usable range of alloys we selected three additional alloys for $T_c$ measurements. 45Cu50Ni has a 100C $T_c$ and has been tested as a viable magnetometer sensor material. The other two, 33Cu58Ni and 39Cu54Ni have $T_c$'s respectively 310C and 250C, while our 380 28% copper alloy, 28Cu60Ni has a $T_c$ exceeding 360C. From the perspective of Curie temperature almost our entire range-under-test is usable as magnetometer sensor material. Alloy 44Cu53Ni is paramagnetic thus out of range.



### 4.4    Initial permeability and coercivity

Our results for these measurements are also consistent with the von Auwers and Neumann (1935) Figures. Our choice of 1% composition increments has enabled a closer look at the location of the ridge of highest permeability/lowest coercivity as noted

by those authors. For 28%Cu compositions 62%Ni gives our highest permeability which could not have been resolved by the earlier data. For 33%Cu and 39%Cu alloys, their nickel contents may need to be slightly higher to achieve maximum permeability and hopefully lower magnetometer noise. Alloy 45Cu50Ni appears to be on target for maximum permeability/minimum coercivity for that copper content. However, by moving only 2% in content to 43Cu52Ni values for both measures significantly improve, permeability increased a factor of six and coercivity declined a factor of 0.75. There may

be a material optimum near that content.

Of particular interest is the very large changes in properties with only single percent changes in contents. Looking at the sequence 42Cu50Ni, 42Cu51Ni, 42Cu52Ni, 43Cu52Ni, 44Cu52Ni, 44Cu53Ni, we note that 42Cu50Ni is a permanent magnet, three later – 43Cu52Ni is near a high permeability optimum, and two after that – 44Cu53Ni is paramagnetic. Von Auwers and Neumann (1935) were aware of a permanent magnetization regime at higher iron contents within these ternary alloys but did

not have a good location for it.

Fig. 10 shows two $B/H$ plots for 42Cu50Ni and 45Cu50Ni, only 3% apart. The first plot shows the squareness and large coercivity of a permanent magnet. The second shows the rounded, low-loss form of our so-far lowest noise magnetometer material.

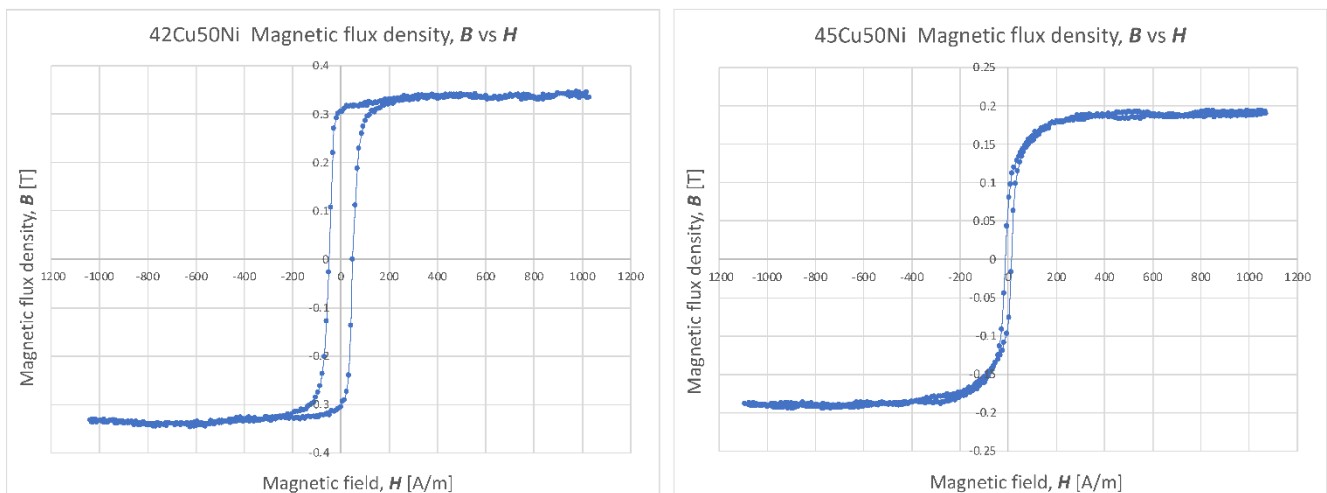

**Fig. 10. Saturation $B/H$ curves for alloys 42Cu50Ni and 45Cu50Ni.**

## 4.5    Fluxgate tests

We expected that as we increased copper content both power consumption and noise PSD would be reduced, the former from the lower field strength needed for saturation and the latter due to the well-known relation between saturation and noise PSD. These relations were both confirmed. What we did not know in advance was how hypothetical poorer magnetic anisotropy levels might impact the measured magnetic properties and noise levels. Narod (2014) predicted that such impacts should vary only slowly with anisotropy and our data confirm that. By far the biggest impact magnetic properties have on sensor noise performance is that of saturation induction.

## 5    Concluding remarks

Our investigation of the 28-45% copper permalloy regime's magnetic properties has led us to alloys which have yielded fluxgate sensors with noise PSD and power consumption improvements over those of the legacy 6Mo81.3Ni permalloy composition. Racetrack sensors of our lowest noise and power alloy, 45Cu50Ni, have noise PSD levels well below 10pT/√Hz at both 1.0Hz and 0.10Hz, easily satisfying the 1-second INTERMAGNET requirement (Turbitt, 2014).

We have now begun to refine our investigation, seeking additional improvements in noise PSD and/or power consumption. Our first alloy selected for further sensor testing, 43Cu52Ni, shares its 100C $T_c$ with our lowest noise alloy, 45Cu50Ni, but has much higher initial permeability. Many uses for low noise magnetic sensors require long durations of data collection, and sensor stability, over both time and temperature is an issue. Our future investigations must address these properties.

Our present results have relied heavily on the existence of the data presented by von Auwers and Neumann (1935), but no such comprehensive examination of molybdenum permalloys was ever undertaken (Chaston, 1936). There may yet be room for improvement of molybdenum permalloys in fluxgate sensors, with molybdenum content higher than that of the legacy 6% materials. In our future work we plan to investigate these alloys.

The performance of these new alloys is expected to enable further miniaturization of the fluxgate sensor while preserving geophysically useful magnetic sensing performance.

## Acknowledgments

That authors wish to thank Christian Hansen, Andrew Schmitt, Matt Miller, and Toby Tompkins for their work manufacturing and machining the permalloy samples, Tino Smith for his work assembling the fluxgate cores, and John Bennest for his work characterising $B/H$ loops.



**Code and Data Availability**

Data and source code used in the creation of this paper can be accessed by contacting the authors.

**Author Contribution**

BBN wrote the manuscript with contributions from DMM. DMM produced the alloy ingots, fabricated, and tested the racetrack fluxgate sensors. BBN produced the magnetic testing specimens, performed the magnetic material testing, and translated von Auwers and Neumann (1935)..

**Competing Interests**

B. Barry Narod operated Narod Geophysics Ltd., which manufactured fluxgate magnetometers until the company ceased
production operation in 2008. David M. Miles and B. Barry Narod hold provisional US patent 63/164,045 related to the use of the described copper alloy regime for magnetic field instruments and magnetic shielding.

**Financial Support**

This research has been supported by the National Aeronautics and Space Administration (grants no. 80NSSC19K0491 and 80GSFC18C0008).

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



## Appendix A: Translation of "On Iron-Nickel-Copper Alloys of high initial permeability" by Otto v. Auwers and Hans Neumann

**Preface to the translation**

In this translation my intent has been to replicate in English the style as it was composed in German. The reader in English will find many of the sentences awkward, long and sometimes using a questionable vocabulary. The inverted order of many sentences is as it appears in German, above and beyond the usual word order differences between English and German.

Much of this would be the writing style of that time, 1935, in both languages. My hope is that I have been able to provide the look and feel of writings of that time.

Translation notes are included in-line. I would like to acknowledge the helpful comments on the translation from F. Harpain and T. O'Connor.

**About the authors:**

Otto von Auwers was born in Heidelberg on July 1, 1885, son of chemist Karl Friedrich von Auwers. He studied natural sciences at the universities in Heidelberg, Munich, and Marburg. He then worked as an assistant at the universities of Gdansk and Greifswald, where he also did his doctorate.

From 1924 he worked as a physicist in the laboratories of Siemens & Halske, where he mainly worked in research. In 1935 he was able to habilitate at the University of Berlin and was employed there in 1943 as an associate professor. After the war ended, Von Auwers went to Clausthal in 1946 as a full professor, where he taught for three years. Von Auwers died on November 4, 1949, at the age of 54.[1]

Hans Neumann worked at Siemens & Halske in Berlin. He is known for the invention of a diaphragm loudspeaker with a planar voice coil (US patent US1987412A, 1935), and for Siemens magnetic alloy, M1040, which from 1934 to 1941 held a record for maximum magnetic permeability[2]

---

[1] Wikipedia, Germany
[2] R. Bozorth, 1951, Ferromagnetism, p. 120.



**Provenance:**

This paper was difficult to locate. This original reprint copy was provided by the Central Laboratory of Wernerwerkes of Siemens & Halske, and made its way to Dr. Egon Ritter von Schweidler, a Vienna physicist. Von Schweidler was vice president (1939-1945) of the Austrian Academy of Sciences.

There is a gap between von Schweidler's death in 1948 and the locating of this photocopy. It was found in a box of loose

materials in 2007, in the Austria Academy of Sciences Library in Vienna, scanned to a pdf file, and forwarded to colleagues at Zentralanstalt für Meteorologie und Geodynamik, (ZAMG) in Vienna, now GeoSphere Austria.

I wish to acknowledge help of the AAS library's director and staff who catalogued the loose material and found the photocopy, and B. Leichter and colleagues at ZAMG who enabled the paper's arrival in my hands.

B. Barry Narod




*Überreicht vom Zentrallaboratorium des Wernerwerks der Siemens & Halske Aktiengesellschaft*

# Über Eisen-Nickel-Kupfer-Legierungen hoher Anfangspermeabilität

# On Iron-Nickel-Copper Alloys of high initial permeability

by

Otto v. Auwers and Hans Neumann

580                                                    with 20 Figures

Communication of the Central Laboratory and the Department for Electrochemistry of "Wernerwerkes" of Siemens & Halske A.G. and the Research Laboratory, Siemensstadt

Received March 25, 1935,

----------------------------

585                 Special printing from Scientific Publications of the Siemens-Factory Volume XIV Issue 2,

Published by Julius Springer, Berlin, 1935 (Not circulated) Printed in Germany

[English translation 2020, B.B. Narod]



## 1. Magnetic properties of high permeability FeNiCu-alloys[3]

Among magnetic materials with higher initial permeability ($\mu_0$) alloys comprise a special type. It is possible with alloys to obtain not only higher but also more uniform $\mu_0$-values than it is with pure metals, for which the variable content of impurities has much larger influence on the magnetic values than with the alloys. For 12 years the $\mu_0$-values of pure and of silicon iron stayed between 400 and 1000. It was therefore a huge step forward for them for the equally important areas of electrical measurement and telecommunications for materials with higher initial permeability and smaller coercive force ($H_c$), which

was achieved in 1923 by G.W. Elmen (1), with the making of an FeNi-alloy ("Permalloy" with 78.5%Ni and 21.5%Fe)[4], with a particular annealing showing 10-times higher $\mu_0$-values (4000 to 10000). Because of the extensive technical applications found in the meantime for the FeNi-alloys, this was the territory of a large number of researchers who have gone in various investigative directions, usually with the technical goal to achieve higher $\mu_0$-values. This was finally made possible by the addition of specific metals to the FeNi-alloys; among these Cr, Mo and especially Cu have proven particularly suitable. A very

well-known magnetic FeNiCu-alloy is Mu-Metal (2), (3), an alloy with 76%Ni, 17%Fe, 5%Cu, 2%Cr, with $\mu_0$-values to 20,000[5]. Other magnetic alloys of the FeNiCu-system that have found uses in the electrical technologies are Thermalloy (5) by J.T. Kinnard and H.T. Faus, an alloy of about 70%Ni, 30%Cu with some Fe, and the very similar Monel-metal (6), a natural alloy with variable composition (about 67%Ni, 29%Cu, balance Fe and other metals). Because of their negative temperature coefficients both alloys were utilized to compensate the temperature coefficients of permanent magnets (as magnetic shunts in

an air gap). Furthermore, should be mentioned those cold-rolled FeNiCu-alloys (namely, about 40%Ni, 60%Fe with Cu-additions to 15%) alloys investigated by O. Dahl, J. Pfaffenberger, H. Sprung (7), M. Kersten (8) and F. Preisach (9), which because of their admittedly low, but very field-strength-independent permeability could win new interest for telecommunications.

---

[3] Chapters 1 and 3 are work of H.Neumann (measurements joint with H.Reinboth), chapter 2 O. v. Auwers.

[4] The statements in this work are weight-percentages.

[5] With an alloy of 72%Ni, 11%Fe, 14%Cu, 3%Mo, H.Neumann (4) has realized the highest ever measured $\mu_0$-value (recently to 51,000).



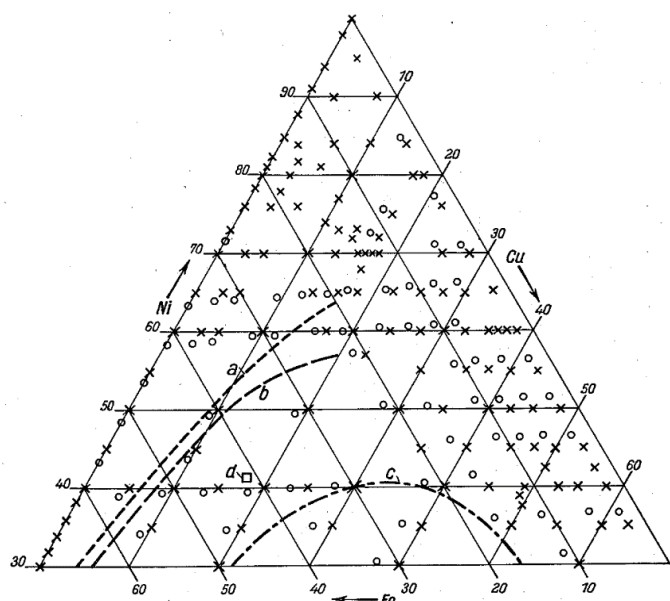

**Fig. A1. Locations of alloys investigated in the FeNiCu-diagram in weight-percent (x) and in atomic-percent (o). Locations of phase separation region boundaries (a, b, c, d) taken from other authors.**

A systematic investigation of the FeNiCu-system is so far still not accomplished. Already R.Vogel (10) has detected that in the FeCu-system there exists a phase separation ["miscibility gap" in the original] region reaching far into the ternary zone (up to about 40% Ni), gradually narrowing (Fig. A1, curve c); this limit applies however only for higher temperatures (about 1000C). P.A. Chevenard (11) et al. concluded from dilatometer measurements, that the phase separation region at room temperature extends into the Nickel corner in the ternary region (Fig. A1 curve b), and O. Dahl and J. Pfaffenberger (12) could essentially confirm this limit by magnetic measurement. Their findings shift the course of the limit of the phase separation region (Fig. A1 curve a) towards still higher Ni content. A. Kussman and B. Scharnow (13) in contrast, with measurements of the coercive force on the line 50%Ni, 50%Fe (to the Cu-corner) put the limit on the phase separation region to about 17%Cu, 42%Ni set (Fig. A1 point d). The FeCu and NiCu binary systems were as well metallurgically and magnetically investigated. For the first system we can name the works of R. Sahmen (14), A. Müller (15), R. Ruer (16), R. Ruer and F. Georens (17), A. Kussmann (18), for the second case are those of W. Guertler and G. Tammann (19) and E.H. Williams (20) et al.

From the above statements, because of the phase separation region, alloys in the FeNiCu-system in the neighbourhood of the FeCu-line are not expected to have high permeability; in the neighbourhood of the NiCu-line, particularly towards the Cu-corner, the Curie-points are too low, and on these grounds we cannot expect very high values of permeability; there remains a quite sizable phase separation zone, adjacent to the highly permeable FeNi-alloys, that few have investigated. The problem of the present investigation was then, to investigate and clarify the FeNiCu-system by measurements of the magnetic properties, and for various heat treatments to ascertain the extent and form of the highly permeable zones in the ternary system. A further



sought verification was whether a connection existed between the height of initial permeability as compared with the coercive

force and the magnetostriction.

**Making of the alloys**

The alloys were cast as ingots from 7 kg melts made in a high-frequency-oven. The starting materials used were Mondnickel of 99.84% purity, wood charcoal iron and electrolytic copper. Compositions were specified as values of the weights. The poured ingots were rolled out to plates of 0.35mm, from which were punched rings of 60mm outer- and 45mm inner diameter.

A final annealing process followed in dry hydrogen.

**Measurement methods**

The measurement method was the usual ballistic ring method, in which about 10 rings were set in ring-formed ebonite casings to keep out mechanical stresses (when direct winding the cores) (21); because of the mainly high $\mu_0$ values, the air-form correction was needed in only a few cases.

What was recorded: the magnetization curves and loops for a maximum magnetic field strength $\boldsymbol{H_{max}} = 10$ Oe; this value suffced, because of the high permeabilities for most alloys, for achieving the limiting values for coercive force ($\boldsymbol{H_c}$) and remanence ($\boldsymbol{B_r}$), and in many cases even for the saturation. For the alloys with larger coercive forces ($\boldsymbol{H_c} > 3$ Oe) the field was increased to 100 Oe; for the magnetostriction measurements sheet metal strips were made, and for control were annealed together with some rings from the same alloy. For these rings the coercive force was measured ballistically, an $\boldsymbol{H_c}$-

measurement for the strips by the familiar ballistic method; An $\boldsymbol{H_c}$ measurement of the strips according to the known ballistic method (in which the sample during the determination of the apparent remanence is pulled from an induction coil), failed because of the too small sample cross section of 3.5 mm² which would have necessitated a much too high winding number of the induction coil, of roughly 100,000 windings. The measurement was made with the astatic compensating magnetometer of H. Gerdien and H. Neumann (22), and in spite of the small apparent remanences (of only 20 Gauss) and the small coercive

forces (as low as 0.02 Oe) the strips still possessed sufficient sensitivity. In this instance the measurement was only possible by having the sample approach very closely to the moving coil magnetometer. For this reason, we did not use the normal two field coils in opposing circuits, rather utilized a long through-spool, the magnetic moment of which is cancelled in a compensating field in our operating moving-coil system (22), upon which the second, astatic moving-coil worked.

**Locations of investigated alloys**

There were about 130 alloys investigated, their locations are given in Fig. A1 with crosses (x). For a number of alloys, the location is also given in atomic-%. The locations of those with (O) indication points, in most cases did not significantly deviate



from the locations of the cross. The coherence of the individual points is clear; for the alloys for which only the percents-by-weight are stated, the atomic-percentages are so close besides, that a draftsman-made diagram was not possible.

## Error sources

The magnetic values for the investigated alloys, to some degree scatter around naturally within a particular neighbourhood. Here the following influences are responsible: fluctuations in the alloy composition, particularly a varying influence from contamination, as well as inevitable fluctuations from rolling and annealing processes. For the analysis of the results also arise errors owing to the occasional inadequate density of the alloy-point locations. In contrast to these errors, the errors of the magnetic measurements of only about 2% are very small. Because in general during repetition of the melting and heat treating

it is possible to obtain a tolerance of only about 10 to 20%, for the quantitative reproducibility of the magnetic values therefore the above accuracy of 2% is entirely satisfactory, in order to obtain with reasonable certainty in a sequence of alloys, for example, the optimum for the initial permeability of a composition. Thus because of the aforementioned scatter in the magnetic values it is difficult to decide if particular small irregularities in the plotted curves have a real underlying reason or not.

## Heat treatments

Since our investigations started with high nickel content, to wit the connection to Permalloys, we first of all retained even during variations of the copper content that heat treatment ("Permalloy treatment", one-hour long heating at 900C with air quenching at 625C) which was designed by G.W. Elmen to obtain for these alloys the best $\mu_0$ values. Comparative measurements with other heat treatments (for example one hour heating at 900C with air quenching or with slow cooling of the oven) did not yield significant deviations with respect to the location in the later described regions of highest initial

permeability below 70%Ni but did of course yield differences in the size of the magnetic values. As a second heat treatment with a principally different effect on the alloy was chosen a two-hour long heating at 1100C with slow oven cooling ("1100C treatment"). The following results thus cover in the main these two heat treatments.



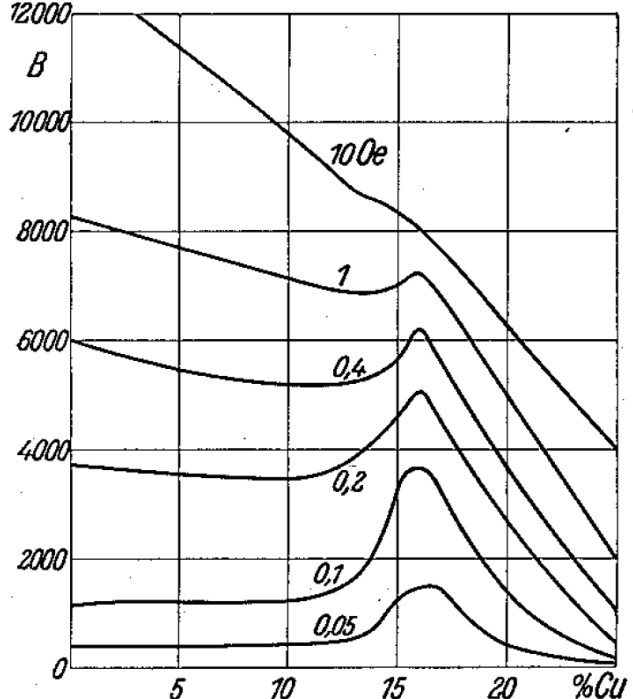

**Fig. A2. Induction *B* as function of Cu-content for various field strengths; section for 70% Ni; Permalloy-handing.**

**Analysis of the measurement details**

The results of the numerous (and unreported) individual measurements were evaluated in the following manner where sections for various Ni-contents (between 40 and 90%Ni) were placed in the alloy triangle, and the magnetic properties, for all $\mu_0$, $H_c$ and the induction $B_s$ for $H = 10$ Oe, were presented as functions of the Cu-contents. For example, Fig. A2 presents for a section of 70%Ni the *B*-value as a function of the Cu-contents, for various field strengths. One recognizes an instability in the magnetic properties for about 16%Cu, for small *H*-values (zone of initial permeability) is particularly prominent and which almost vanishes with increasing saturation. Fig. A3 presents a similar typical section for 40%Ni. Here is present a very pronounced maximum in magnetizability for 50%Cu, which is separated from still higher magnetizable alloys with small Cu-content by a wide valley between 30 and 49%Cu. This valley is due to the channelling through of the edge of the upper peak of the phase separation region, a zone of poor magnetizability. Similar curves result, when one selects the sections for constant Cu-content, whereas for constant Fe-content the differences were not so obvious. Since according to both of the last images the magnetization for small field strengths, that is, $\mu_0$, is not only of technical significance, but also constitutes a very characteristic property, therefore one could dispense with the presentation of the field dependence of induction; hence in Fig. A4 the $\mu_0$-values for the Permalloy-treatment are presented as functions of the Cu-contents, and the various Ni-contents presented as a parameter. One recognizes clearly that there is always a quite peaked maximum found in the initial permeability,



for each of the various Cu-contents examined. Surprisingly furthermore is the fact, that even with lower Ni-contents (45%Ni) and very high Cu contents (50% Cu) high $\mu_0$-values (4000) were still found. There exists therefore a certainly narrow, but quite long zone of highest $\mu_0$-values in the FeNiCu-system between 40 and 80%Ni, and within this narrow zone is a $\mu_0$-maximum definitely present at about 70%Ni. Figure A5 gives the same measurements for the 1100C-treatment; here also there is a narrow zone of higher initial permeability with a $\mu_0$ maximum present, only this [maximum] is shifted towards smaller

Ni-contents (60 to 65%Ni). The coercive forces yield, processed in the same manner, give the same plot, equally so for the maximum permeabilities. The location of the high permeability zone becomes more apparent if in "van't Hoffschen" triangles one puts in lines of constant magnetic properties; so in Figures A6 through A13 are drawn the lines of constant values for $\mu_o$, $\boldsymbol{H}_c$ and $\boldsymbol{B}_s$ for the two heat treatments. Figures A7 and A9 are photographs of spatial models of the initial permeability ($\mu_0$-values presented vertical) for the pair of heat treatments, which make these conditions very apparent. For the individual Figures

we observe the following: Figs. A6 and A7: For the Permalloy treatment the zone of highest initial permeability takes a narrow, band-form from the Permalloy-alloys with 78.5%Ni ($\mu_0 = {\sim}8000$) with steady rise of $\mu_0$-values to its top, peaking from 68 to 75%Ni and 9 to 19%Cu, a maximum with $\mu_0 = 12{,}000$ and then sloped relatively steeply (the saddle minimum for 60%Ni is probably not real), sagging to 52%Cu with $\mu_0 = 500$ to 1000, the values for technical iron. Also drawn in the diagram are the phase separation regions as reported by various authors; one sees that its impact is still noticeable up to the highest Ni-

contents beneath the $\mu_0$-maximum for 70%Ni, as a flaring of the lines of constant initial permeability. In the lower part of the highly permeable zone around 40%Ni, 50%Cu the band of higher permeability bends somewhat in the direction towards the alloy 50%Fe, 50%Cu. Figures A8 and A9 give similar ratios for the 1100C-handling. Here the location for the $\mu_0$-zone is the same, only the $\mu_0$-maximum within this zone has shifted to lower Ni- and higher Cu-contents. Whereas the direction the $\mu_0$-zones between $\mu_0 = 4000$ and $\mu_0 = 12{,}000$ is roughly the same as for the Permalloy-handling, for this heat treatment the

highest initial permeability on the NiFe-line is for more than 78.5%Ni, but less than 88%Ni. The influence of the phase separation region on the form of lines of constant permeability is only indicated for alloys of more than 30%Cu; the saddle minimum for 60%Ni, 30%Cu, 10%Fe as well as the bent ends on the $\mu_0$-zones that went towards the FeCu-line are completely absent here. Also noteworthy is that the width of the $\mu_0$-comb has become noticeably smaller, as follows from the denser positioning of the contours' layers. This could be because the chosen higher treatment temperature and slow cooling effect a

precipitation hardening itself much more noticeable than with the Permalloy-handling with the lower anneal temperature and the air quenching. Figures A10 and A11 show the $\boldsymbol{H}_c$-values for the two heat treatments. The small $\boldsymbol{H}_c$-zone developed quite similarly as that of the initial permeability, and this was true for both heat treatments. For the Permalloy-treatment the $\boldsymbol{H}_c$-minimum moves towards the alloy 81%Ni, the $\mu_0$-maximum towards 78.5%, whereas for the 1100C-treatment the valley of coercive force bends towards 88%Ni, much as in Fig. A8 the initial permeability does for the same heat treatment. The $\mu_0$-

and $\boldsymbol{H}_c$-zones thus develop similarly for each of the two heat treatments, but of course differently for the two heat treatments. Again, one notices a degree of influence of the phase separation region; it is worthy of mention here that the 1100C treatment



produced a larger zone of smaller $H_c$-values (0.03) as compared to the Permalloy treatment in which the $H_c$-minimum is shifted towards smaller Ni and higher Cu-contents. There is nevertheless this fact to take into consideration, that because of the very small remanences in the neighbourhood of the NiCu-line the coercive force is difficult to determine there, this is in

contrast to $\mu_0$, the measurement of which presents difficulties only as one approaches very small values. Therefore, one could think of circumstances that for 1100C treatment (Fig. A11) the site of the $H_c$-minima within the small zone is also the same as the site of the $\mu_0$-maxima in Fig. A8. With that having been shown, the 1100C-handling delivered an unambiguous, definite site for the optimum of both magnetic values ($\mu_0$ and $H_c$) for about 60% Ni, 28% Cu, whereas for the Permalloy-handling the main effect is exerted only on the initial permeability; while the locations of $\mu_0$-maxima have shifted in the direction of the

Permalloy-alloys (for about 68 to 75%Ni), probably the locations of $H_c$-minima have gone in the opposite direction, or otherwise expressed: For the 1100C-handling the sites of optimums for $H_c$ and $\mu_0$ stayed the same, for the Permalloy-handling in comparison both optimum zones moved apart. To decide if this effect is a result of the differing anneal temperatures or of the cooling rate, a series of alloys was annealed one hour at the same temperature of 900C, in this series some alloys were again oven cooled, and some were quenched in air. The result was as follows: the location of $\mu_0$-maxima within the zone of

high permeability alloys remained practically the same for both the 900C heat treatment and for the Permalloy treatment, so that the determining factor was not the size of the cooling rate but is credited to the height the anneal temperature. The difference between both heat treatments for 900C consists thusly, that the width of the $\mu_0$-ridges for slow cooling is smaller than for the air-cooled, particularly for low Ni- and high Cu- contents. The explanation for this might therefore again be sought in the differing actions of precipitation-hardening in the neighbourhood of the mixed phase zone, which for slow cooling must

become stronger than for quenching in which the precipitation is suppressed; therefore, the zone of higher permeability, that is, smaller coercive force, must become constricted with increased precipitation-hardening.

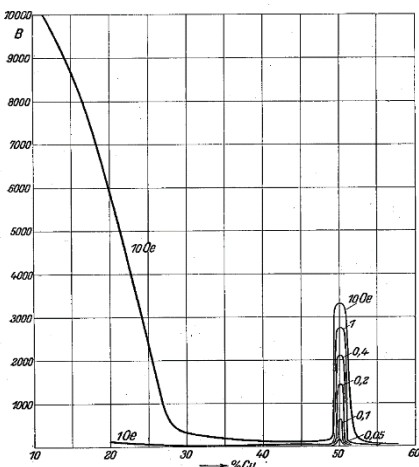

**Fig. A3. Induction _B_ as function of Cu-content for various field strengths; section for 40% Ni; Permalloy-handling.**





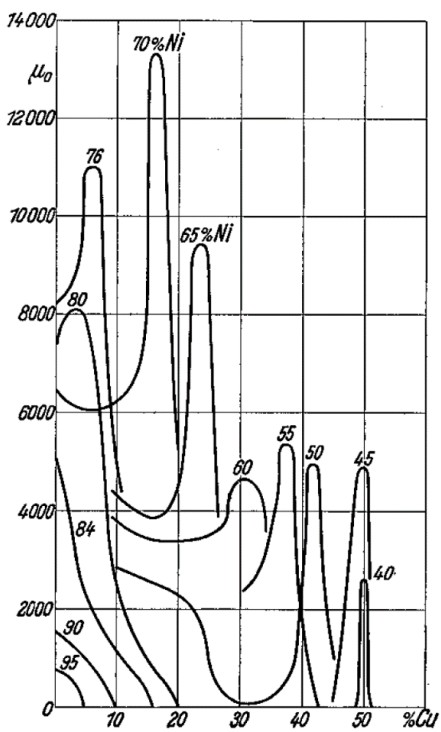

Fig. A4. Initial permeability $\mu_0$ as function of Cu-content; sections for various Ni-contents; Permalloy-handling.

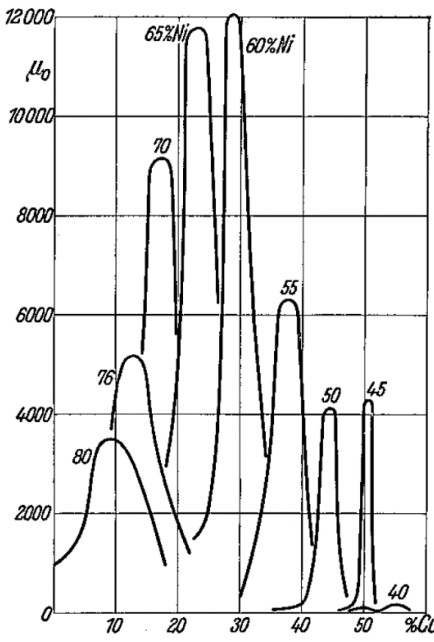

Fig. A5. Initial permeability $\mu_0$ as function of Cu-content; sections for various Ni-contents; 1100C-handling.



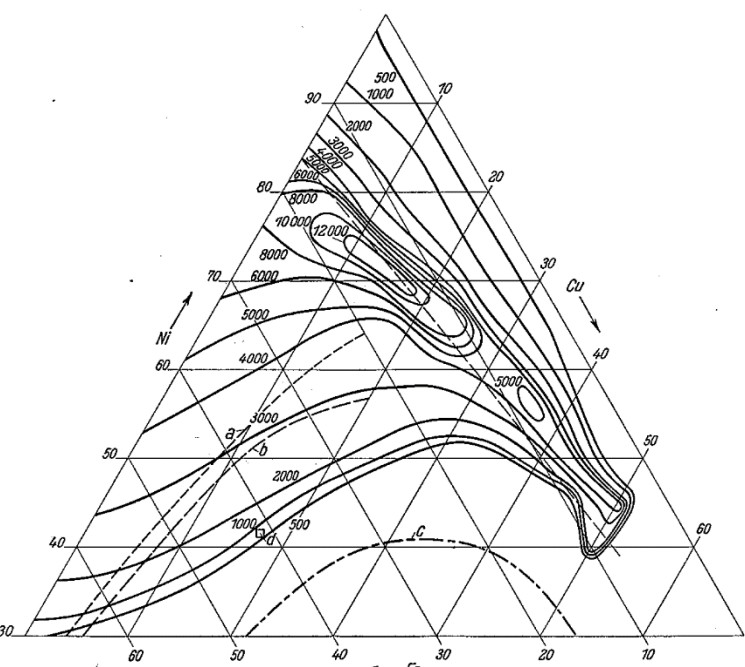

**Fig. A6: Lines of constant initial permeability (-) in the FeNiCu system; Permalloy handling; Boundaries of phase separation region from other authors (a,b,c,d); Line of constant Ni:Fe ratio ( ---).**

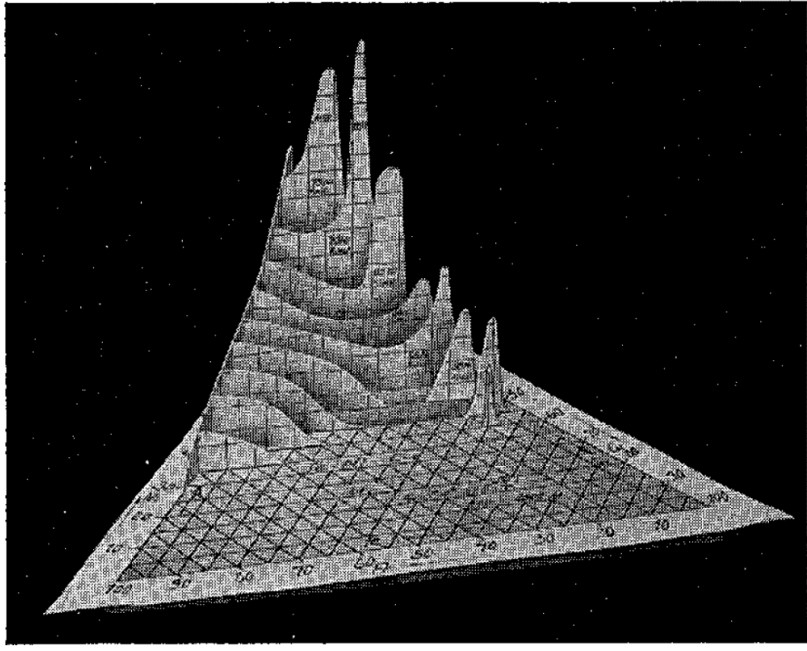

**Fig. A7: Spatial drawing of Fig. 6; ordinate = initial permeability.**




Figures A12 and A13 give the lines of constant induction at $H = 10$ Oe, for the two heat treatments (Permalloy and 1100C treatment). Here the locations for the $\mu_0$-contours shown for lower Ni-contents (about 40%Ni) are only implied, which perhaps alloys might leave for the same magnitude of magnetic field strength of 10 Oe, the values for the higher permeability alloys (for higher Ni content) having gone to saturation, for alloys with smaller Ni contents having not. Possibly the proximity of the phase separation region is responsible for the shapes of the curves, which shapes are still not exactly known for room-temperature in this zone. Apart from this, the locations of the contour lines for the two heat treatments are nearly the same, as the height of the saturation values depends only a little on the heat treatment - irrespective of the formation of an ordered atomic distribution; in contrast it is known that the ease of the magnetization, that is the field strength for a specified induction (that is, the permeability), depends strongly on the heat treatment.

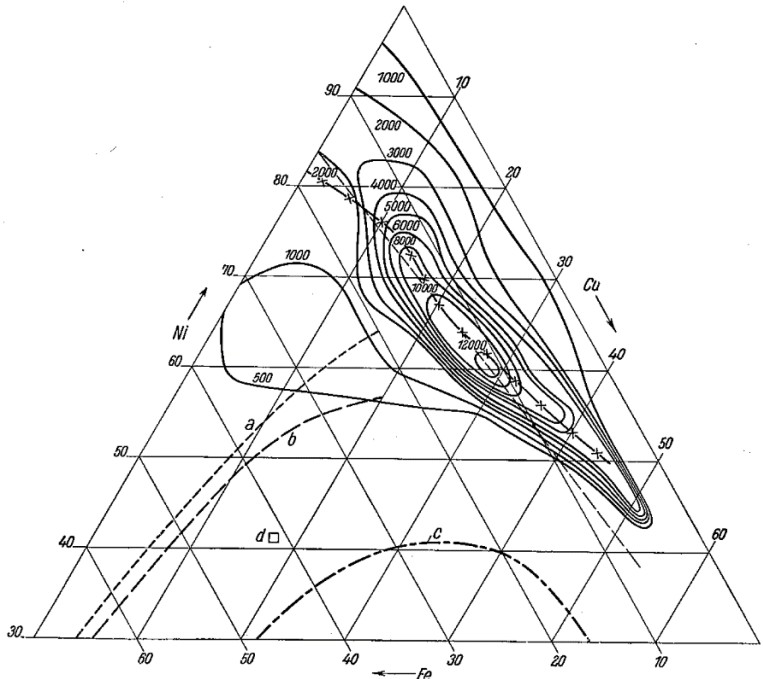

**Fig. A8: Lines of constant initial permeability (-) and line for magnetostriction null in the FeNiCu system; (1100C-handling)(-x-x); Line of constant Ni:Fe ratio ( ---); Boundaries of phase separation region from other authors (a,b,c,d).**





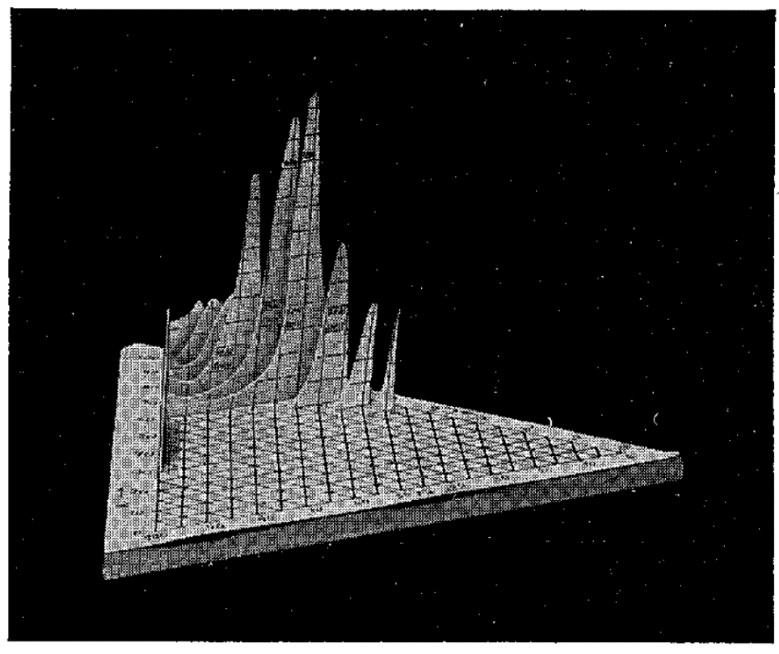


**Fig. A9: Spatial drawing of Fig. A8; ordinate = initial permeability.**

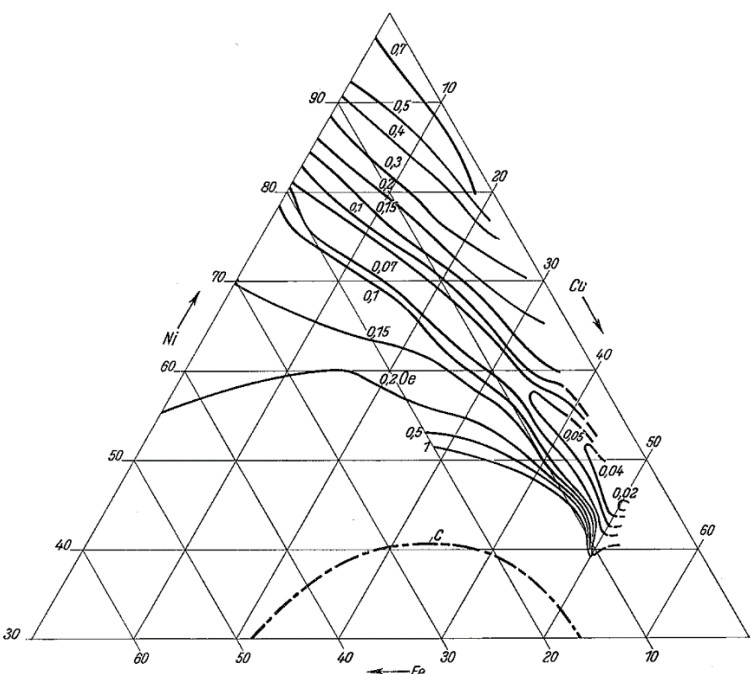

**Fig. A10. Lines of constant coercivity (-) in the FeNiCu system; Permalloy-handling.**



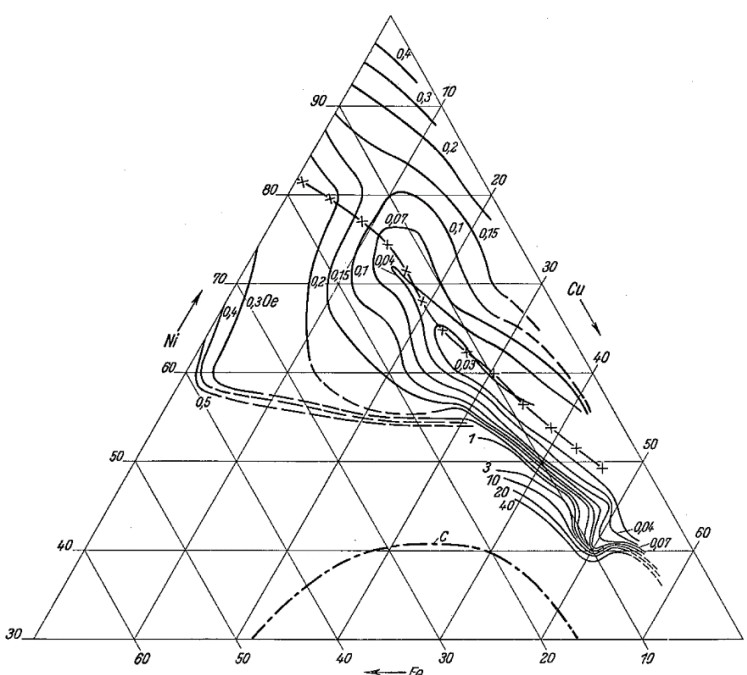

**Fig. A11. Lines of constant coercivity (-) and line of zero magnetostriction in the FeNiCu system (-x-x); 1100C-handling.**

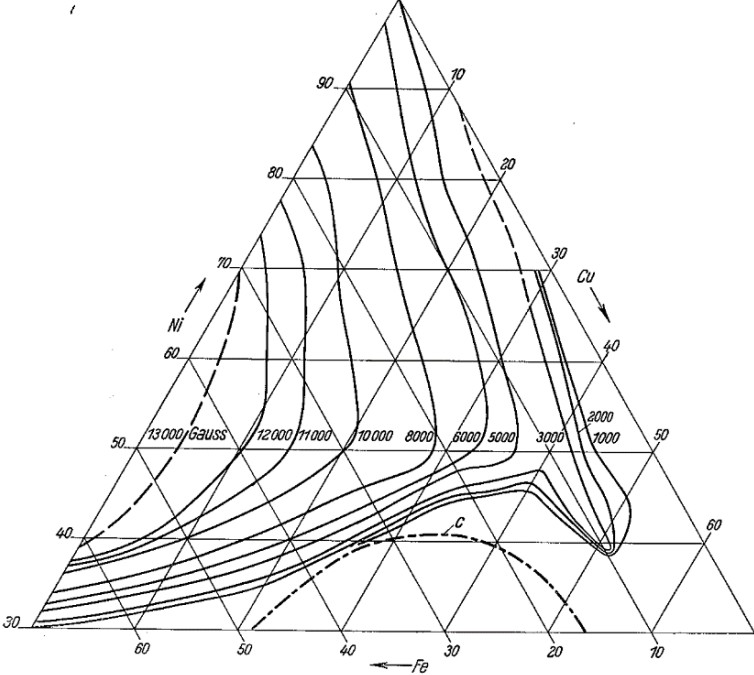

**Fig. A12. Lines of constant induction (in Gauss) (-) for $H = 10$ Oe in the FeNiCu system; Permalloy-handling.**




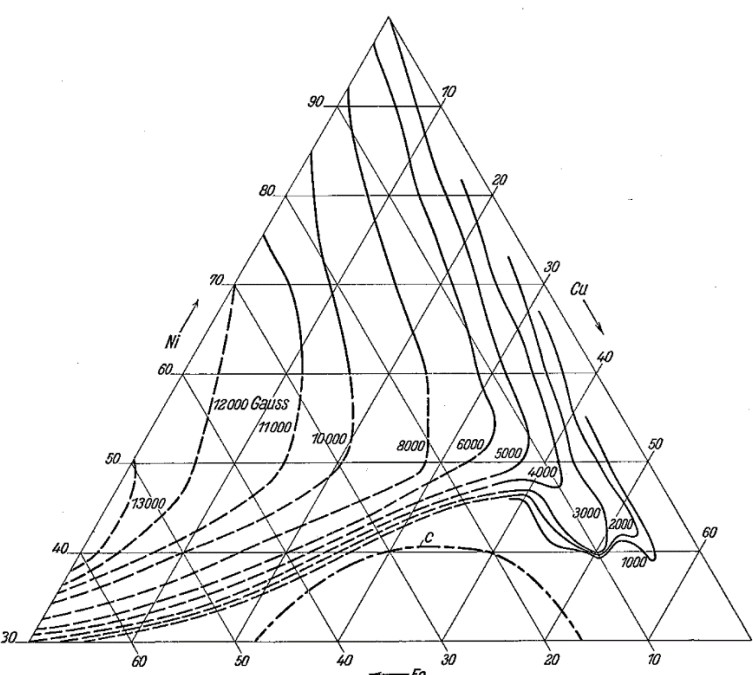

**Fig. A13. Lines of constant induction (in Gauss) (-) for $H = 10$ Oe in the FeNiCu system; 1100C-handling.**

**Changing the heat treatment**

This becomes especially clear when one follows the trace of $\mu_0$ or of $H_c$ along the highly permeable zones. Thus, because of the different positions of the optimal values of $\mu_0$ and $H_c$, in every case there has to be a boundary at which both heat treatments yield the same values for $\mu_0$ and for $H_c$. Figure A14 presents each of the achieved maximum values of $\mu_0$ for each of the heat treatments as a function of the Cu content while Fig. A15 gives the same for the $H_c$ values. While the associated Ni-contents

for both curves are not entirely the same, it is clearly apparent that for $\mu_0$ the boundary is at 22%Cu and for $H_c$ the boundary is at 13%Cu; the reason being that the contour lines of the $\mu_0$ crests are at that place coincidentally parallel to the lines of constant Cu content as is apparent from Figs. A6, A8 and A11. This is only one quantitative indication for the already mentioned fact, that for high Ni and low Cu-contents the Permalloy treatment gives the better magnetic values, while for low Ni- and high Cu-contents it is the 1100C-handling.

**Reproducibility of the measurements on sheet metal strips**

Some evidence is shown in Fig. A16, for the good reproducibility of measurements regarding the locations of $\mu_0$-maxima c.f. the $H_c$-minima, for the section 70%Ni and 1100C-treatment. $\mu_0$ and $H_c$ are presented as functions of the Cu-content, once for the sheet metal rings used and once for the later-on treated sheet metal strips, on which the associated magnetostriction measurements were done. The two large drawn-in circles represent the $\mu_0$-values of the rings that were annealed together with





the sheet metal strips; the conformity between the values and the curves is, as shown, very good with respect to the positions of the optima, even though regarding the magnitude of the magnetic values there are present, as previously mentioned, somewhat larger fluctuations.

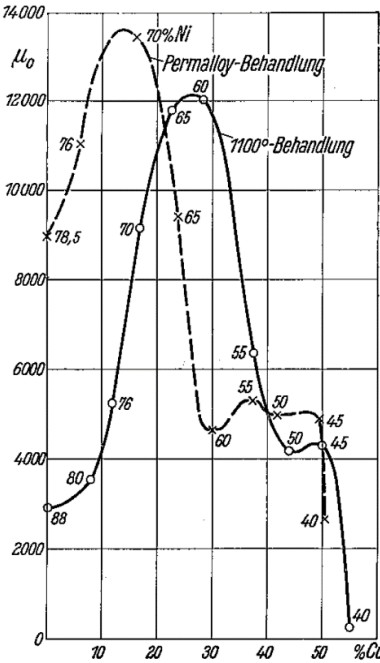

**Fig. A14. Points of $\mu_0$-maxima as function of Cu-content (the figures on the curves signify the Ni-content).**



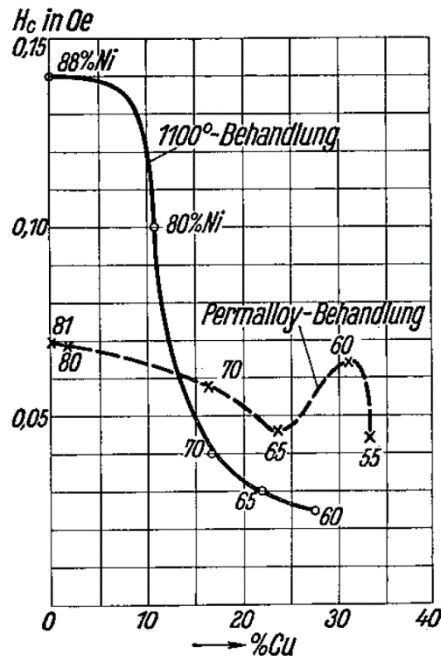


**Fig. A15. Points of minimum coercive force $H_c$, as function of Cu-content (the figures on the curves signify the Ni-content).**

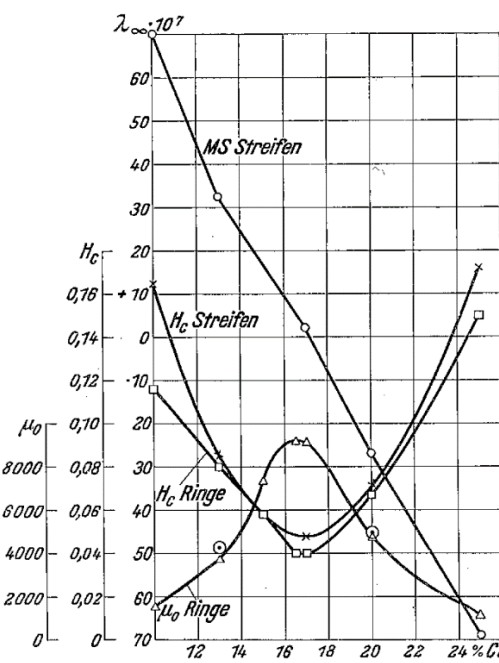

**Fig. A16. Initial permeability $\mu_0$ and coercive force $H_c$ of rings; coercive force $H_c$ and saturation magnetostriction ($\lambda_\infty$) MS on strips as function of Cu-content; section for 70% Ni; 1100C-handling.**



## 2. The Connection with Magnetostriction

The idea to bring a connection between high initial permeabilities and small magnetostriction-values $\lambda$, was first mentioned by L.W. McKeehan (23) in the form of a stress-hypothesis with respect to the then highest known permeability values for Permalloys, and found wider development subsequently in the works of N.S. Akulov (24), R. Becker (25) and R. Becker and M. Kersten (26), which led M. Kersten (l.c.) to the simple formula

$$X_0 = \frac{2T_\infty^2}{9\lambda_\infty \sigma_I} \tag{1}$$

($X_\infty$ = initial susceptibility, $T_\infty$ = magnetic saturation, $\lambda_\infty$= saturation-magnetostriction, $s_i$ = mean stress). In broad outline the usefulness of this formula (after replacement of the difficult to obtain measurement for inside tension $s_i$ by $\lambda E$)

$$\mu_{0max} = \frac{8T_\infty^2}{9\pi E\lambda^2} \tag{2}$$

($E$ = elastic modulus) was confirmed in the FeNi-series, by M. Kersten (27).

In this situation it appears interesting[6] if and which connections exist for the above described high-permeability alloys of the FeNiCu-system with the magnetostriction values of these alloys. Thus 28 alloys of this system were investigated for their longitudinal magnetostriction. According to the earlier work of O. v. Auwers (28), R. Becker (29) and M. Kornetzki (30) one can from their documents also determine the behaviour of volume-magnetostriction.

The related alloys were the same as those used in the first part the work. For the magnetostriction measurements metal strips

of 100 x 10 x 0.35 mm³, together with some rings that had been used for the magnetic measurements, underwent the same (c.f. Fig. A16) heat treatments (1100C-treatment and Permalloy-treatment). The present investigation was then done on those alloys slowly cooled from 1100C. The measuring apparatus was recreated as that of M. Kornetzki (30), the measuring-length 30mm.

### A. Longitudinal Magnetostriction

All alloys were measured in fields up to 1000 Oe. This field-strength proved to be adequate to determine with sufficient

accuracy, in spite of the unfavourable aspect ratios, not only the saturation magnetostriction but also the linear increase of

---

[6] Prior search in this direction was done in Inst. of Prof. R.Becker, Charlottenburg, in 1933 by M.Kornetzki and H.Neumann.



longitudinal magnetostriction, for all the high permeability alloys with exception of the alloys with 40%Ni. From this gradient one can, as is well known (31), determine the volume-effect, since[7]

$$\alpha = \frac{\Delta V/V}{H - H_0} = 3\frac{\Delta l}{l} \tag{3}$$

The validity of these relations was confirmed by M. Kornetzki (l.c.) for iron and by O. v. Auwers[8] in numerous FeNi-single-
crystals from F.Lichtenberger (32). Fig. A17 may give an example of the magnetostriction-values for alloys with 70%Ni, which fundamentally shows the recurring behaviour for all other section tests. From these curves, for which the connection with the magnetization intensity $B$ will be elaborated later, were determined the magnetostriction-saturation values $\lambda_\infty$, that means the $\lambda$-values, for which the $\lambda$-$H$-curves passed into the linear part above the technical saturation together with the associated field strengths $H_0$. With the thus collected $\lambda_\infty$-values one can plot various sections, for which Fig. A16 gives an
example. The intersections these curves make with the zero-line give the chemical compositions, for which the integrated[9] magnetostriction of the polycrystalline alloys becomes zero. Carrying these values into the alloy-triangle, its connecting-line (Fig. A18, curve a) (with exception higher Nickel-contents) becomes practically coincident with the contour lines of initial permeability for the same heat treatment (1100C) (Fig. A18, curve b) within the error-bounds.

Figure A18 presents other lines of constant saturation magnetostriction values, which in their aggregate course show a close
connection with initial permeability (Fig. A8). In Figs. A8 and A11 the lines for magnetostriction null are drawn in as well; here also the connection is clearly recognized. Thus is given convincing evidence not only for the far reaching importance of Equations (1) and (2), but at the same time a plausible explanation for the course of initial permeability in the FeNiCu-system: the initial permeability has the maximum value, where the longitudinal magnetostriction was zero.

---

[7] on the impact of $H_c$ cf. below and v.Auwers (28).
[8] Unpublished.
[9] This naturally says nothing about the magnetostrictive behaviour in the various crystal-orientations, which individually can be different from zero and even need not agree in polarity; c.f. F.Lichtenberger (l.c.)





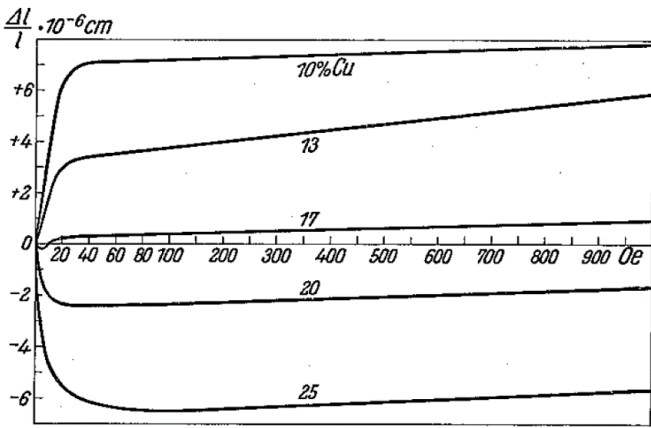

**Fig. A17. Magnetostriction $\Delta l/l$ as function of the applied field strength for 70% Ni, for various Cu-contents; 1100C-handling.**

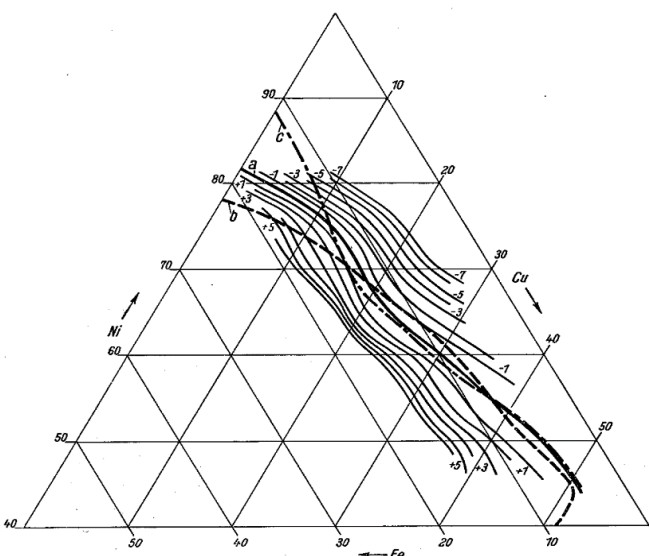

**Fig. A18. Lines of constant magnetostriction; line of magnetostriction nulls (curve a); Line of highest initial permeability for 1100C-handling (curve c) and for Permalloy-handling (curve b).**

Against the obvious objections, that for Equations (1) and (2) for $\lambda = 0$ their validity must fail, one can exercise two points of

view: 1. the denominator could be assembled as the sum of two[10] energies, for which the second in general can have negligible

size, as compared to $\lambda_\infty * s_i$, whereby the value $\mu_0 = \infty$ is excluded when $\lambda$ goes to zero; and 2, one must consider, that the

---

[10] The magnetostrictive strain-energy in the crystal lattice and as always - also in a well annealed lattice - available deformation-
energy.





equations are derived for "anisotropic" magnetostriction, that is, by neglecting the fact that magnetostriction is generally to some extent anisotropic, that means it does not go to zero simultaneously in all orientations of the crystal lattice.

That alone is sufficient to always maintain for the gross magnetostriction strain in polycrystalline materials a certain underlying stress level that originates in the crystallographic anisotropy of magnetostriction and must formally make itself noticeable as an additive component in the denominator.

## B. The $H_0$ Field

Taking from the $\lambda - H$ curves the outer ones, depending on the shape [11], for which field-values $H_0$ for saturation-magnetostriction $\lambda_\infty$ have been reached, and contour plotting in the alloy-triangle (Fig. A19), one can also recognize here a
close connection of the course of the crest-lines with the initial permeability peaks: the contour lines close to minima of coercivity and the saddle between them. But maybe this connection is less a dependence on the magnetostrictive properties and is rather an immediate effect of the regime of permeability, as $H_0$ must naturally be smaller, as ever lighter magnetic saturation values were reached.

## C. Volume magnetostriction

In the same manner were handled the linear ramps of the $\lambda - H$-curves (cf. Fig. A17) above technical saturation. They gave, as already observed, [p.103 (sic)], again one-third the volume effects. Notice once again the lines of constant volume-effect $\Delta V/3V$ in the alloy-triangle, as follows in Fig. A20; its comparison with the course of both crest-lines of initial permeability, in spite of their larger complexity can yet still reveal some coherence. In this example the crest lines of initial permeability for both heat treatments go with great accuracy across the narrowest and lowest position of the canyon between 65 and 75%Ni.
The alloys with 40%Ni given up to 1000 Oe still easily bent $\lambda - H$-curves. A basis for their ill form might be sought in the larger coercive forces of these alloys (cf. Fig. A11). It is interesting, that the volume magnetostriction in the FeNiCu-system is similar to that of the FeNiCo-system[12], a similar embedded saddle was found in it.

---

[11] The other dimensions were the same for all probes, so that the values "ceteris paribus" [all else being equal] are comparable. The internal field-strength oriented $H_0$-value might be very much smaller.
[12] cf. O. v. Auwers; 1, e, p.827, Fig. 6.



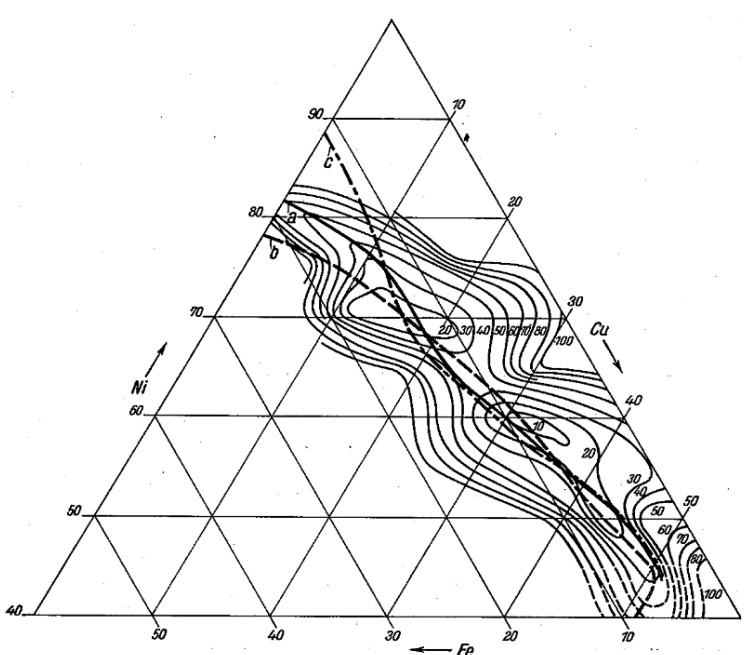

**Fig. A19. Lines of constant field strength for saturation magnetostriction; Line of magnetostriction null (curve a); Lines of highest initial permeability for 1100C-handling (curve c) and for Permalloy-handling (curve b).**

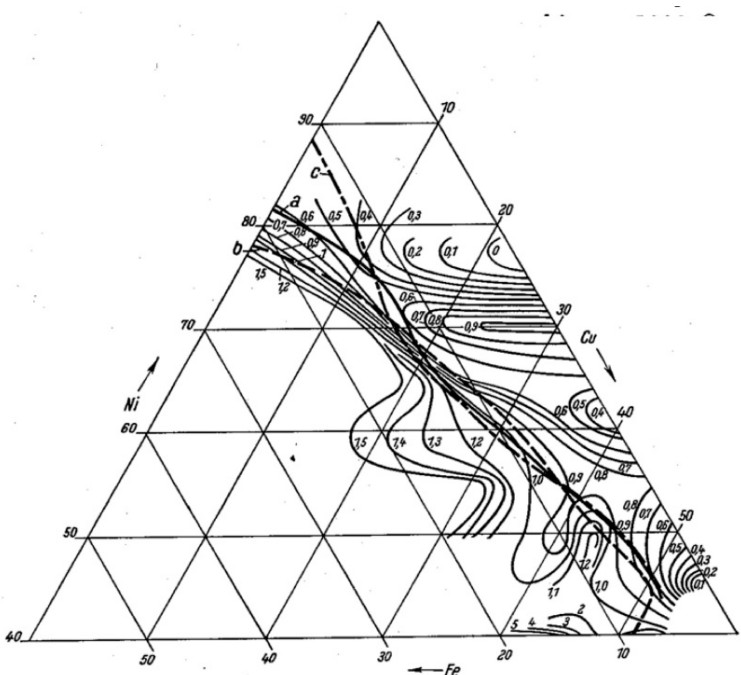

**Fig. A20. Lines of constant volume magnetostriction, (x $10^{-9}$); Line of zero magnetostriction (curve a); line of maximum initial permeability for 1100C-handling (curve c) and for Permalloy-handling (curve b).**



**D. Effects of thermal handling**

The magnetostriction measurements described above were all for alloys for which 1100C slow cooling were executed. To clarify the question, considering the above-noted effects of thermal processing on the initial permeability, and whether it may also make a noticeable effect in the magnetostriction values, some samples made as metal bands were given Permalloy handing. It was thus shown that the heat treatment has no noticeable effect on the null line of longitudinal magnetostriction. More about this and of a likely equivalent effect on volume magnetostriction will be reported fully in a later publication.

**E. Results**

In summary one can say that the present magnetostriction measurements on high permeability FeNiCu-alloys not only narrows the relation between initial permeability and the magnetostrictive behaviour these alloys have expressed, but together have given a convincing story for the accuracy of modern theories of magnetization in ferromagnetic crystal lattices, theories that establish all of the legendary importance of magnetostriction for the magnetic behaviour of ferromagnetics. This connection

has still one evident restriction, in so far as the tests point, that aside from magnetostriction as already indicated above, still other magnitudes can have significance for the permeability (cf. for example alloys above 70%Ni). This can also be expressed, that the position of the magnetostriction curves, in contrast to the curve of maximum permeability, is entirely independent from the thermal process used.

**3. Overview**

In an overview of the entirety of the acquired results, the following can be said: There was found a long, narrow zone of highest $\mu_0$-values, below 70%Ni found practically coincident with the zone of smallest coercivity. In the zone above 70%Ni the heat treatment had a particular influence and occurs on both magnetic attributes ($\mu_0$ and $H_c$) in the same sense. For both magnetic values there existed within the narrow zone an optimum zone, for $\mu_0$ a maximum, for $H_c$ a minimum; for an 1100C anneal temperature both optima fall together, whereas for a 900C anneal temperature the $\mu_0$ maximum has shifted to 70%Ni and the

$H_c$ minimum to 45%Ni. On the width of the high permeability zone and its form taken in the lower part, there is a phase separation region extending from the FeCu-line, and due to differences in the segregation processes the particular heat treatment has some influence.

The line for zero magnetostriction in polycrystals with less than 70%Ni, practically coincided with the zone of $H_c$ minima and $\mu_0$ maxima; the sites of $\mu_0$ maxima and $H_c$ minima respectively, within the high permeability zone had no natural expression

in the line for zero magnetostriction. In the region above 70%Ni there exists a deviation of the magnetostriction line away from the regime of optimal $\mu_0$ and $H_c$ zones respectively; the line of zero magnetostriction for both heat treatments runs

approximately to the alloy 82%Ni, 18%Fe, then for highest initial permeability for 1100C-handling to 88%Ni, 12%Fe, for the Permalloy handling to 78.5%Ni, while the $H_c$ minimum for Permalloy-handling went to 81%Ni and for the 1100C-handling to 88%Ni. About the as yet unexplained reason for these deviations, more should also be reported.

In addition to the clearly confirmed connection with the magnetostriction there is the following remarkable fact: the direction of the narrow high permeability zone, originating at the FeNi-line, passes towards the Cu-corner, while the location of the exit point of the regions, as discussed above, can shift according to the heat treatment. The whole regime of zones of higher permeability can - irrespective of the deviations in the vicinity of the FeNi-line - indicate, as whether we are involved in this zone with alloys of constant Ni:Fe ratio. One draws a specific line originating at 84%Ni 16%Fe (respectively the ratio Ni5:Fe)

and going towards the Cu-corner, which follows, especially from about 70%Ni, along the general regime of $\mu_0$-crests (Fig. A6) and also lies together, in good agreement with the line for the magnetostriction null (Fig. A8). The divergences for lower Ni-content and within the large zone near the NiCu-line should perhaps be taken conditionally; the alloys in this zone have typically very low Curie-points, for example under 100C for an alloy of 45%Ni, 50%Cu, 5%Fe, compared with an alloy of 70%Ni, 10%Cu, 20%Fe, which has 490C. Because of the various intervals of the measurement temperature (+20C) to the

Curie-points the results for the individual alloys were thus not directly comparable, and it is conceivable that with lower temperatures the alloys with low Curie-points give values to the zone of higher permeability, something others could also do. This question is to be investigated later.

**Summary.**

There was found in the FeNiCu-system a zone of high initial permeability and small coercivity, its location goes from about

80%Ni, 20%Fe reaching down to 40%Ni, 50%Cu, 10%Fe.

For the large, upper part of this zone there existed a broad connection with magnetostriction in such a way that at alloys with highest initial permeability the magnetostriction sign reversed.

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

965    ----------------------------------------------------------------------------

Printed by Spamer A-G., in Leipzig.