# Peer review of "Copper permalloys for fluxgate magnetometer sensors"

_EGUsphere, 2023_

## Author Response (AR1)

RC1: 'Comment on egusphere-2023-2191', Anonymous Referee #1, 30 Oct 2023

This work is a continuation of the work "On Iron-Nickel-Copper Alloys of high initial permeability" by Otto v. Auwers and Hans Neumann. The presented work is devoted to the search for the optimal composition of the Cu-Ni-Fe alloy for the fluxgate sensor. The search for the optimal composition took place in the region of the largest values of the initial magnetic permeability in the direction of decreasing saturation induction. The authors found the alloy composition - 45mass%Cu 50mass%Ni and 5 mass%Fe (45Cu50Ni), the core of which had a small coercive force of 4.5A/m and the lowest saturation induction of 0.2T. At the same time, the Curie temperature is not very low = 100°C, which makes it possible to use this alloy in a fluxgate sensor. It turned out that the fluxgate made of this alloy has the lowest values of noise power spectral densities at 1.0Hz and 0.10Hz - 6 and 8 pT/√Hz, respectively, and the lowest values of power consumption - 50mW. It was found that the greatest influence of magnetic properties on the noise characteristics of the sensor is induced by saturation.

It is shown that the noise power spectral densities and power consumption of the fluxgate sensor from the new 45Cu50Ni alloy have lower values than those from the 6.0-81.3 Mo Permalloy alloy. It would also be interesting to see a comparison of their magnetic properties.

The paper fully corresponds the requirements of the journal and can be recommended for publication after minor changes:

In the sentences (appendix 283) "Four cells presented with red lettering codes, namely 2860, 3558, 3954 and 4550 are our specimens also used in Curie temperature tests and in fluxgate sensor builds." the value 3558 should be replaced by 3358.

Citation: https://doi.org/10.5194/egusphere-2023-2191-RC1

AC1: 'Reply on RC1', David Miles, 23 Nov 2023   reply

Thank you for the review. Great catch regarding the "3558" label! We have updated the manuscript accordingly.

Citation: https://doi.org/10.5194/egusphere-2023-2191-AC1

**RC2**: 'Comment on egusphere-2023-2191', Anonymous Referee #2, 03 Nov 2023

The abstract of the history of permalloy development is great! Based on these historical papers the authors present results with a new or rediscovered alloy with copper contribution for application for fluxgate magnetometers.

I truly appreciate the effort in manufacturing and testing of permalloy materials for fluxgate magnetometers. Particularly because all the necessary development steps are nowadays in a commercialised world extremely difficult to manage.

Thus the paper is absolutely worth to be published.

I have only one minor comments. The general conclusion, that the material with copper contribution has lower noise compared to molybdenum alloy, is not true. Maybe it is true in case you compare it with cores made by molybdenum alloy in your own lab, however, Fe-81-Ni-6Mo cores made by other groups have very similar noise properties as the ones you made by alloy with copper contribution. The noise floor between 2 and 10pT/sqrt(Hz) at 1Hz, depending on core size and excitation power, seems to be attainable with different alloy compositions.

Citation: https://doi.org/10.5194/egusphere-2023-2191-RC2

**AC2**: 'Reply on RC2', David Miles, 23 Nov 2023

Thank you for the review. It is correct that we were comparing our 45Cu50Ni results with 6Mo81Ni results of our own builds, and with the legacy 6-81 Infinetics ring cores. In Section 3.6 we have now limited our comparisons to only those.

Since this preprint went public new historical information has come to light, regarding the production of ring-cores, both in Germany and USA. We have made changes in Section 1.2.6 to reflect the new information.

In particular, in the USA the Hamilton Watch Company was indeed involved in producing in quantity the NOL 6-81 alloy and foil. In Germany in the 1990's ring-cores were produced with noise levels "less than 2 pT/rtHz in the frequency band between 0.1 and 64 Hz." (Fornacon et al, 1999). Text to reflect this is added to our paper, and the reference is also added. Also, the ring-cores developed at that time were a cooperative effort of Karl Heinz Fornacon (Technischen Universität, Braunschweig), Manfred Müller (Zentralinstitut für Werkstoffforschung,Technischen Universität Dresden) and Yuri V. Afanasiev (Russian Academy of Sciences, Moscow, and elsewhere). This work was also undertaken in a part by Halbzeugwerk Auerhammer, now Auerhammer Metallwerk.

New text is added to reflect all this new (to us) information, cited as personal communications.

We have added this reference:

Fornacon, K.H., Auster, H.U., Georgescu, E., Baumjohann, W., Glassmeier, K.H., Haerendel, G., Rustenbach, J., Dunlop, M.: The magnetic field experiment onboard Equator-S and its scientific possibilities, Ann. Geophysicae 17, 1521-1527, https://doi.org/10.1007/s00585-999-1521-3 1999.

Citation: https://doi.org/10.5194/egusphere-2023-2191-AC2

List of all the lines with edits:

191-202

288

351-352

365

473-475

41-42, 60, 207, 506 and Afanassiev